# Urbanity and the dynamics of language shift in Galicia

Mariamo Mussa Juane[1], Luis F. Seoane[2,3], Alberto P. Muñuzuri [1] & Jorge Mira [4]

Sociolinguistic phenomena often involve interactions across different scales and result in social and linguistic changes that can be tracked over time. Here, we focus on the dynamics of language shift in Galicia, a bilingual community in northwest Spain. Using historical data on Galician and Spanish speakers, we show that the rate at which shift dynamics unfold correlates inversely with the internal complexity of a region (approximated by the proportion of urban area). Less complex areas converge faster to steady states, while more complex ones sustain transitory dynamics longer. We further explore the contextual relevance of each region within the network of regions that constitute Galicia. The network is observed to sustain or reverse the dynamic rates. This model can introduce a competition between the internal complexity of a region and its contextual relevance in the network. Harnessing these sociodynamic features may prove useful in policy making to limit conflicts.

[1] Group of Nonlinear Physics. Departamento de Física de Partículas, Universidade de Santiago de Compostela, 15782 Santiago de Compostela, Spain.
[2] ICREA-Complex Systems Lab, UPF-PRBB. Dr Aiguader 88, 08003 Barcelona, Spain. [3] Institute Evolutionary Biology, UPF-CSIC, Pg Maritim Barceloneta 37, 08003 Barcelona, Spain. [4] Departamento de Física Aplicada, Universidade de Santiago de Compostela, 15782 Santiago de Compostela, Spain. Correspondence and requests for materials should be addressed to A.P.M. (email: alberto.perez.munuzuri@usc.es)

Hobbes's Leviathan[1] is an example of social contract theory according to which people partly submit their autonomy to a broader institution. Such a 'social contract' has never been signed. Rather, evolutionary social dynamics adjusted the parts involved, effectively shaping relationships between individuals[2,3]. Similarly, before institutions took hold, precise mathematical laws emerge out of free choices and interactions between people[2,4–6]. Tensions often ensue between individual choices and coarse-grained dynamics. Similar tensions ripple across scales of social organization. As globalization proceeds, it becomes urgent to understand these tensions between organization scales: Are such clashes unavoidable? Are their outcomes always the same? How are their paces regulated? What aspects of the interactions between individuals constrain such phenomena?

We study competing scales of organization through a relevant sociolinguistic scenario. Language is at the center of human activity. Its variety seems threatened by globalization as numerous tongues may disappear soon, replaced by hegemonic ones[7–9]. In the long run, Axelrod's minimal model of culture dissemination[10] predicts one of two exclusive scenarios: either (i) small, isolated communities maintain global heterogeneity; or (ii) a well-connected, homogeneous culture displaces all others. The transition between both extremes is harsh, leaving no intermediate options. Axelrod's model shows mathematical conditions for the prevalence of each regime. We wish to contribute to the understanding of similar segregation vs. homogenization scenarios and their unfolding in time.

More precisely, we examine language shift—the process whereby a vernacular tongue gets replaced. This could be enhanced by globalization, which brings together different languages and might precipitate a choice between them. Intuitively, these shift dynamics could be faster in well-connected environments (e.g., cities, which boost technological, economic, and other human interactions[11–13]) than in sparsely connected ones (e.g., rural areas). Accordingly, homogenizing dynamics could be faster in cities. Our data here show that, counterintuitively, the opposite in fact happens. Language shift evolves more slowly in cities, while rural areas reach their equilibria faster. We explore two mechanisms giving rise to different paces of these dynamics in urban and rural setups: one based on internal complexity and another on contextual relevance of each region within a broader context.

Our exhaustive data analysis comes from the Autonomous Region of Galicia (northwest Spain), where Galician (a Romance language close to Portuguese) and Castilian (Spanish) are co-official. This region offers a unique laboratory to undertake studies similar to the one presented here. Despite covering 5.9% of the Spanish territory (29,575 km² out of 505,370 km²) and containing 5.8% of the Spanish population (2.72 of 46.56 million), by 2015 Galicia encompassed 30,244 Singular Population Entities (SPEs)—a whopping 49% of the Spanish total 61,695[14]. Briefly, SPEs are villages, towns, or cities. See Supplementary Information for details on the definition of SPEs. This spectacular atomization means, for example, that over 27,000 Galician SPEs had less than 100 inhabitants by 2016[14]. Relatively large cities (∼300,000 inhabitants) also exist.

We combine the empirical time series of fractions of Galician, Spanish, and bilingual speakers with an analytic model based on differential equations[15,16]. Thus, we quantify perceived language prestige levels, similarity, and (crucially here) the rate at which the dynamics unfold. First, we conduct our analysis on a series of Galician regions independently—i.e., assuming that all information needed to reproduce the data is internal to each region. This is insightful, but in reality, Galician regions influence each other. Hence, we complete the model turning the independent regions into nodes of a network. Both approaches offer

mechanisms to explain different paces for rural and urban dynamics—as observed in the data. One explanation hinges on each region's internal complexity. The other one relates to that region's relevance within a wider network. These mechanisms spring from different organizational scales. Numerical analyses show how both mechanisms compete as these organizational scales interfere. All these results are discussed in the following section. In the Discussion, we overview possible consequences for policy making and other real-world scenarios. Similar effects across organizational scales could ensue from general globalization scenarios, conferring more value to our insights.

## Results

**Effect of internal complexity.** Both Galician and Spanish are Romance languages sharing co-official status in Galicia (northwest Spain) since the 1978 Spanish Constitution. An abundant literature exists studying their coexistence from perspectives, including national identity, historical issues, and trends of Galician language use[17,18]. We adopt a new angle by using differential equations to describe the time series of use of both languages. Available data of Galician, Spanish, and bilingual speakers spans back, at most, to the first third of the 20th century. Extrapolating backward suggests that a generalized contact started effectively with the 20th century[15,19].

Our approach builds upon numerous recent developments that model language shift with differential equations[15–29]. This field was largely sparked by the Abrams–Strogatz (AS) model[22] which successfully accounts for the decline of 42 real-world minority languages in contact with hegemonic counterparts. A simple differential equation captures the likelihood that a speaker would change her tongue depending on (i) the fraction of speakers of the other option and (ii) perceived language prestige (encoded by a parameter $s$ inferred from the empirical data). Solutions of these equations do not allow mixed stable populations, hence predicting the extinction of one tongue in the long term. The AS model dealt with two exclusive languages $X$ and $Y$, whose monolingual speakers compose the population with fractions $x$ and $y$ ($x + y = 1$), respectively. We use an extension of the AS model introduced by Mira and Paredes[15]. This allows a third option, B, of bilinguals that make up a fraction $b$ of the population (such that $x + y + b = 1$). The model equations read:

$$\frac{dx}{dt} = c[(1-x)(1-k)\,sx^a - x(1-s)(1-x)^a],$$
$$\frac{dy}{dt} = c[(1-y)\,ksy^a - ys(1-y)^a]. \tag{1}$$

This model is detailed in the Methods section, where historically related works are briefly discussed. The stability of Eq. (1) has been well studied[6,19,30,31] and evaluations of this model against empirical data (including the Galician case) exist[15,19,32]. We build upon these results. At this point it is worth mentioning that the research on language maintenance and shift has evolved from the classical studies of Fishman[33], including new perspectives from mathematical approaches[34]. For example, our model yields a phase space that allows, on the basis of parameters calculated from fits to empirical data, for the quantitative measure of the degree of risk of the weaker language[16,19,30,31]. The model dynamics depend on two initial conditions ($x(t = t_0)$ and $y(t = t_0)$), i.e. the initial distribution of speakers) and four parameters ($a$, $c$, $k$, and $s$). Of these, $k$ (termed *interlinguistic similarity*) controls access to the bilingual group; $a$, $c$, and $s$ are inherited from the AS model. These parameters are discussed in the Methods section or, as they become relevant, below.

Using time series derived from recent polls[35], we looked at 20 different Galician regions which we label $i = 1,...,20$ throughout the paper (see Supplementary Table 1). Fitting our model

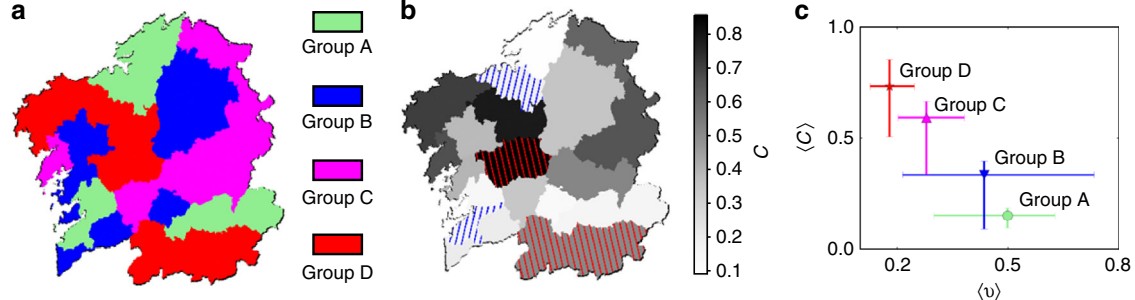

**Fig. 1** Diversity of sociolinguistic dynamics in Galicia. **a** Galician regions colored according to the steady states that the dynamics of speakers should reach according to model Eqs. (1 and 4) and our data. **b** Parameter $c_i$ measured for each Galician region coded in grayscale (see color code at the right). Blue stripes tilted rightwards mark regions that are prominently urban, while red stripes tilted leftwards mark those regions which are prominently rural. **c** Parameter $c_i$ correlates inversely with the percentage of urban population as defined in the main text. The average of both $c_i$ and $u_i$ has been taken for each group of regions (A to D). Error bars indicate the maximum and minimum values for each group

equations to this data, we obtain estimates of Galician and Spanish prestige ($s_i^G \equiv s_i$, respectively $s_i^C \equiv 1 - s_i$) and other model parameters ($a_i$, $c_i$, and $k_i$). In Methods, we explain how the steady states of the model depend mostly on $s_i$ and $k_i$ (see ref. [16] for details). These equilibria, which we note ($x_i(t \to \infty)$, $y_i(t \to \infty)$, $b_i(t \to \infty)$), are predictions about the long-term evolution of Galician and Spanish speakers in each individual region. These results carry the (provisional) assumption that regions evolve independently of each other. Based on the equilibrium that each region attains, we classify them in four qualitatively different groups (Fig. 1a): (i) areas tending to become monolingual Spanish ($y_i(t \to \infty) > (x_i(t \to \infty)$, $b_i(t \to \infty))$; from now on, group A); (ii) regions tending to become mostly bilingual, with Spanish preponderance ($b_i(t \to \infty) > (x_i(t \to \infty)$, $y_i(t \to \infty))$, $y_i(t \to \infty) > x_i(t \to \infty)$; group B); (iii) regions tending to become mostly bilingual, with Galician preponderance ($b_i(t \to \infty) > (x_i(t \to \infty)$, $y_i(t \to \infty))$, $x_i(t \to \infty) > y_i(t \to \infty)$; group C); and (iv) areas tending to become Galician monolingual ($x_i(t \to \infty) > (y_i(t \to \infty)$, $b_i(t \to \infty))$; group D). Such a heterogeneous outcome, intuitively, makes sense provided the diverse, fragmented reality of Galician SPEs discussed above.

These results depend on $s_i$, $k_i$, and the initial conditions within each region. As discussed in the Methods section, the parameters $c_i$ do not affect the asymptotic steady states. Instead, the $c_i$ reflect rates of change observed on these temporal series for each region (Fig. 1b). Note that, while steady states could change in the future due to unexpected socio-political developments that the model cannot capture, the rate at which these time-series have evolved is a more intrinsic property of our current data.

Figure 1b shows the $c_i$ measured for different Galician regions. As we approached this work, we expected rural areas to tend more slowly to their stable states, while cities would decay faster. The rationale for this was that interactions are more frequent in urban setups, and sparser in rural areas. Our model equations assume that, within each region, every speaker interacts with each other. Hence, more frequent contacts, we thought, might correlate with faster dynamics—similar to how chemical reactions do.

Our data contradict this hypothesis. In Fig. 1b we highlighted two notably rural regions (red stripes tilted rightwards, e.g., the mountainous region at the south) and the areas hosting the two largest Galician cities (blue stripes tilted leftwards). This plot suggests that the studied social dynamics unfold faster in rural areas. In Fig. 1c, we group together regions within each of the groups defined above (A through D, Fig. 1a) and plot the group average $c_i$ versus the average proportion of urban population $u_i$ within each group. To compute each region's $u_i$, we scored the number of people in that region living on SPEs of more than 5000

inhabitants and divided this number by the total population in that region (5000 inhabitants was chosen based on a local regulatory threshold, see Supplementary Information for details). Figure 1c shows that more urban population (which would imply more frequent interactions), correlates with slower language shift.

In the literature, we meet a first potential candidate to explain this behavior. Toivonen et al.[36] simulate opinion dynamics in a network. Nodes represent agents with an opinion or state. At every iteration, states are updated stochastically depending on the proportion of neighboring agents with either of the two opinions. The probabilities that determine how opinions evolve in[36] are similar to the ones that enter our Eq. (1). It is also possible to coarse-grain the whole network from the simulations in[36] and write down differential equations (similar to Eq. (1)) that tell how average opinions evolve. Our equations are related to those in ref. [36]—both describe opinion dynamics, and both have been used to study language shift[15,16,19,28]. We chose Eq. (1) because they better account for bilingualism, including the possibility of stable bilingual communities.

Simulations in ref. [36] show how opinion dynamics have a shorter relaxation time in simpler communities, while more complex ones take longer to reach their stable states. Eq. (1) ignore the internal social structure within each region. This, nonetheless, affects the empirical data. Hence, a possible explanation for the results in Fig. 1c is that urban areas have greater internal complexity than rural ones—suggesting that our data is an empirical validation of the results in ref. [36]. We propose that, since the internal network structure is reflected by dynamics of varying speed, the coarse-graining of this internal complexity is captured by different $c_i$ values. We argue that urban environments are more complex given the larger diversity of activities and people, while inhabitants of rural areas are exposed to less varied interactions. It is hence plausible that the proportion of urban area partly captures the internal complexity of each region's intrinsic interaction network.

**Network and contextual relevance.** In the preceding analysis an important factor was missing: the different regions are not independent, but rather, they influence each other. Each region, thus, becomes a node in a Galicia-wide network across which linguistic preferences of different regions affect those of others. In real systems, neighboring areas interact at their borders, or bigger cities could exert a pressure on far away speakers to shift languages. Also, large cities attract trends from other areas, potentially being affected by foraneous linguistic choices to a greater extent. To incorporate this into the model, we modified Eq. (1)

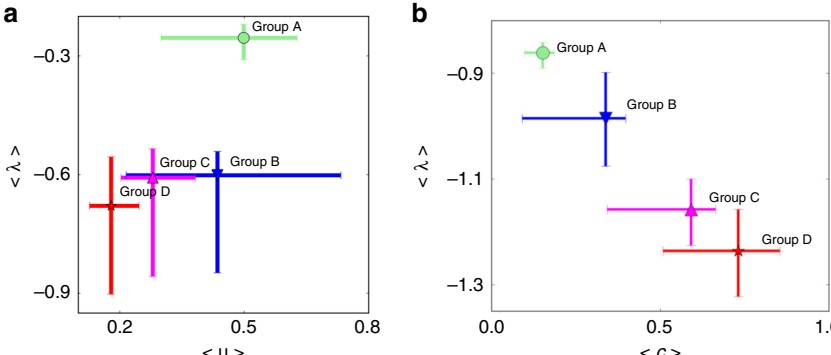

**Fig. 2** Dynamics induced by network structure. **a** Average growth factor <λ> versus coupling percentage of urban population $u_i$ with $\alpha = 0.1$, $\beta = 0.2$. **b** Average linear growth factor <λ> versus <c>. Note that all average values are done within each of the four different groups (A to D). Error bars indicate the maximum and minimum values for each group in Fig. 1a

into the simplest ones that include some interaction across regions[37]:

$$\frac{dx_i}{dt} = f(x_i, y_i) + K_i \frac{1}{N} \sum_{j=1}^{N} \left( x_j - x_i \right),$$

$$\frac{dy_i}{dt} = g(x_i, y_i) + K_i \frac{1}{N} \sum_{j=1}^{N} \left( y_j - y_i \right). \tag{2}$$

Network nodes (i.e., Galician regions) are still labeled by $i = 1, ...,$ $N = 20$. Variables $x_i$ and $y_i$ still represent fractions of monolingual Galician and Spanish speakers within each region. Note the existence of bilinguals $b_i = 1 - x_i - y_i$ such that population remains always normalized within regions. (See Supplementary Information). Functions $f(x_i, y_i)$ and $g(x_i, y_i)$ introduce the same nonlinear interactions from Eq. (1) (see Methods), such that if $K_i = 0 \forall i$ we recover the original model. If $K_i \neq 0$ differences in fractions of speakers in distant nodes exert influences across the network. However, this is not a model of speaker migration—rather of how linguistic choices propagate system-wide. Eq. (2) are equivalent to having a virtual, mean-field node that pulls together the pressure of each linguistic group across all regions. The constants $K_i$ tell us how relevant this mean-field context is for each node. This model is the simplest one with certain network structure. We could have opted for more complex ones (see Supplementary Information), but this naïve, coarse-grained approach already allows us to extract some insight about social dynamics from a systemic viewpoint.

As noted before, the parameters $c_i$ capture how quickly each node reaches its stable state in the original Eq. (1). In the previous section, we showed how these parameters correlate with a measure of each region's intrinsic internal complexity. We wish to put aside this effect now, and study instead the influence of the contextual relevance of each node within the network. From original data, each region has its own value $c_i$. To isolate the effect of the contextual relevance, we set all $c_i = 1$ in all subsequent numerical explorations. This way, the internal dynamics have the same relevance for all nodes, and we control their relationship with the large-scale network through a single parameter for each region ($K_i$). For larger $K_i$, interactions between nodes will be dominant over each region's internal dynamics. In the simulations that follow, each node still retains the measured values of $a_i$, $k_i$, and $s_i$.

Different approaches can be considered to study Eq. (2). Using linear stability analysis (see Methods) the growth factor, $\lambda_i$, for each node is calculated[38]. This measures how quickly a single node evolves into a new stationary state after being perturbed. Note that the physical meaning of the growth factor is similar to

the $c_i$ of the original model, in that both measure the pace at which each node decays towards its steady state.

The dynamics of this extended model easily incorporates nodes that relax to their stable state at different speeds. As it was shown in Fig. 1c, the evolution rates ($c_i$) correlate with the percentage of urban population in each node ($u_i$). To introduce a certain dependence of the $K_i$ on urban population, we try the simple Ansatz: $K_i = \alpha(1 + \beta u_i)$. According to this, each node has a baseline connectivity $\alpha > 0$ to the rest of the network, which is complemented by a factor $\beta u_i$ proportional to the node's urban population. Note how $\beta > 0$ represents that larger cities have a greater capacity to access communications and mass media and, thus, to feel other nodes, which makes intuitive sense. In Fig. 2a we plot the case $\alpha = 0.1$, $\beta = 0.2$. This might roughly correspond with the Galician situation (low general connectivity and relatively low difference between the urban and rural nodes, as it will be explained in more detail below). The growth factor for each node has been calculated. To simplify the visualization of the results, we show the average growth factor $\lambda_i$ across Galician regions included in each of the four groups (A to D introduced above) versus the average $u_i$ in each group. The $\lambda_i$ are negative, so a region decays more slowly the closer its growth factor is to 0 (i.e., the smaller $|\lambda_i|$).

Figure 2a shows that the network structure can induce different convergence speeds for each one of the nodes, and that we can recover a profile of dynamical rates similar to the one obtained with the individual models for each region. Compare the trend shown in this figure with that of Fig. 1c, where group D (mostly composed of rural areas) has the highest decaying velocity (i.e., again, largest $|\lambda_i|$), meaning that regions in group D will relax to their steady state faster than others. Figure 2b shows the growth factor of this illustrative model versus the fitted values of $c_i$. Since both $\lambda_i$ and $c_i$ play the role of convergence time scales for each node, this figure shows how the observed pattern of faster convergence in rural areas can stem either from the internal complexity or from a distribution of contextual relevance for different nodes depending on the proportion of urban population.

**Competing mechanisms.** Hence, we have two competing mechanisms that can induce different convergence speeds to sociolinguistic dynamics across regions: A first one, inspired by the analysis of opinion dynamics in microscopic agent models[36], postulates (i) the existence of a varying internal complexity across Galician regions, and (ii) that this complexity gets coarse-grained (to some extent) into the $c_i$ of the original model. The second mechanism illustrates how the simplest network structure connecting Galician regions can cause rural regions to converge faster to their steady state, just as in our data.

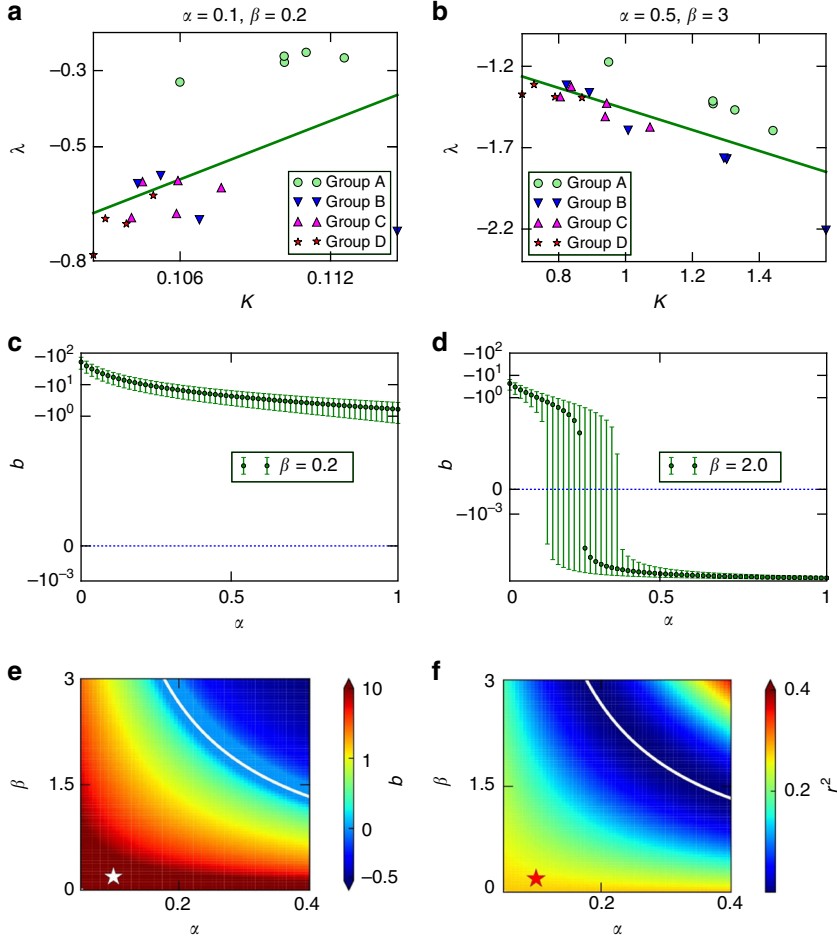

**Fig. 3** Growth factor λ versus coupling parameter K. Positive (**a**) and negative (**b**) slopes in the regression lines indicate faster or slower dynamics for either rural or urban areas. **c**, **d** Variation of the slope $b$ of the regression lines as a function of $\beta$ for two different values of $\beta$, error bars are calculated directly from a linear regression of the data in (**a**, **b**). **e** Color-coded values of $b$ across the $\alpha$ vs. $\beta$ phase diagram. **f** Regression coefficients across the $\alpha$-$\beta$ space diagram. Star marks the location of Galicia (with $\alpha = 0.1$, $\beta = 0.2$)

We seek now to examine more systematically the rates of social dynamics that Eq. (2) can enforce for different $\alpha$ and $\beta$. Figure 3a shows $\lambda_i$ versus $u_i$ for the same values of $\alpha$ and $\beta$ as in Fig. 2a in which rural regions decay faster due to the network structure. Note how fitting a simple regression ($\lambda_i = a + bK_i$) returns a line with positive slope ($b > 0$). In turn, in Fig. 3b, other $\alpha$ and $\beta$ values lead to a different network structure that enforces a faster decay in urban areas, which would result in a linear regression ($\lambda_i = a + bK_i$) with negative slope ($b < 0$). We computed the growth factor for each node for different values of $\alpha \in [0,1]$ and $\beta \in [0, 3]$, and fitted each $\lambda_i$ versus $u_i$ plot to a regression line. Figure 3c and d show the slope $b$ for two fixed values of $\alpha$ varying $\beta$. Figure 3e shows $b$ (color coded) as a function of $\alpha$ and $\beta$. Note that $b > 0$ corresponds to faster rural dynamics and $b < 0$ to faster urban dynamics. This space shows that the model accommodates both these behaviors as a smooth function of $\alpha$ and $\beta$. Urban areas decay slower to their steady state when $\beta$ is smaller, but at some combination of large $\alpha$ and $\beta$ (top-right corner in Fig. 3e) the behavior is inverted and social dynamics speed up in urban areas. Figure 4 illustrates how such a process could play out in Galicia as connection through a mean-field node becomes more prominent.

Figure 3f uses the correlation coefficient $r^2$ to tell us where in the $\alpha$ vs. $\beta$ space this correlation is relevant. Note that $r^2$ here is not primarily a goodness of fit test of $\lambda_i$ vs. $K_i$ (since fitting is not our main goal here), but an index of how this behavior (the pace of the unfolding dynamics) correlates with the percentage of

urban area within different regions. We observe that $r^2$ vanishes in a curve across the $\alpha$ vs. $\beta$ space, just as the slope changes its sign. Also, for low $\alpha$ the network effects should vanish. Since all $c_i$ were set equal to one in the network simulations, different paces for $\alpha \sim 0$ stem mainly from differences in the proportion of speakers of each region. If different $c_i$ were allowed, their effects would become manifest in this low $\alpha$ regime, in which network effects vanish and, instead, internal complexity would determine the speed of unfolding social dynamics. As the context of regions in the network becomes more relevant (by increasing $\alpha$, $\beta$), both mechanisms come into competition because they have different effects on the speed of the dynamics depending on whether a region is more or less urbanized (which, once again, we take as a proxy for internal complexity of that region).

## Discussion

Sociolinguistics (the study of the influence that society has on the way language is used and the society's effect on language) is incorporating the influence of globalization. In this context, Blommaert[39] argues that the world has not become a village, but a complex web of villages, towns, and different types of settlements, all connected by material and symbolic ties, whose effects are not known; a complexity that needs to be examined and understood. Other works[40] have stated that classic sociolinguistics not only ill-fits non-western cases, but it is also difficult to apply to large,

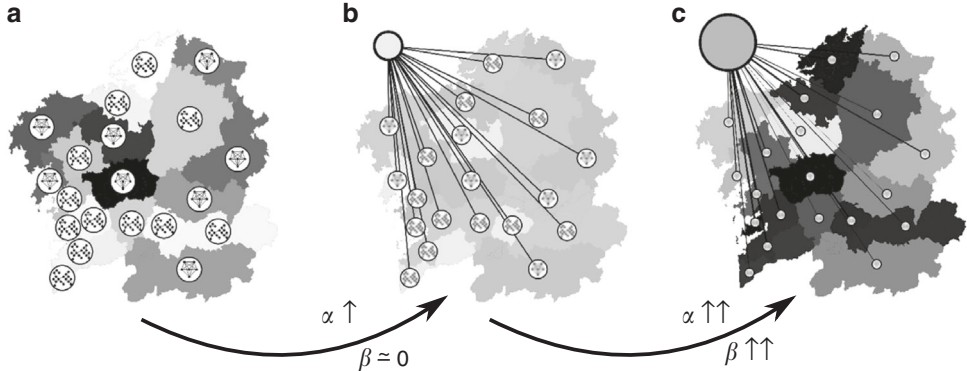

**Fig. 4** Schematic representation of dynamic rates as the coupling of urban areas to a mean-field node is made predominant. **a** Within the proposed model, for low values of both $\alpha$ and $\beta$, speakers within each region evolve mostly independently following its own internal dynamics. Our data indicates that prominently urban regions evolve slower than rather rural regions in this regime. This is in accordance with computational results from ref. [35], who predict slower convergence time for more complex networks. **b** Allowing nodes to interact (in our model, through a mean-field node) diffuses this effect. **c** If connectivity to the mean-field node favors more urban areas (which makes intuitive sense, given the ability of urban areas to collect and cast net-wide trends), our model predicts that prominently urban areas should evolve faster when compared to rather rural areas—i.e., connectivity at a mesoscale could change the seed trends observed when internal dynamics predominate

western urban areas today, arguing that the urban/rural division within a state pertains to a more fundamental distinction, one between two worlds. It is the aim of uncovering such mechanisms that have also guided the present work.

A previous approach to the dynamics of language shift puts forward a model with three relevant parameters ($a$, $k$, and $s$; termed volatility, interlinguistic similarity, and prestige[6]) which, together with the initial conditions, determine whether (i) only one monolingual group survives or (ii) a stable coexistence (also involving bilinguals) is possible[15,16,19]. A final model parameter, $c$, measures the speed at which such social dynamics unfold. This simple scaling factor does not affect the system equilibria and was thus overlooked in the literature.

Here, we have paid attention to the pace of these language shift dynamics and how it compares across Galician regions. The parameter $c$ correlates inversely with the proportion of urban population, $u$, within each Galician region (Fig. 1c). This indicates that our dynamics relax faster to their equilibrium in the rural, and slower in urban regions. We discuss two mechanistic explanations as to why social dynamics should progress at different paces in different regions: One mechanism depends on the internal complexity within a region and another one emerges out of the systemic properties of a Galicia-wide network. Both mechanisms are insightful and suggest strategies for regulators who, in the face of current globalizing trends, might prefer to slow down or speed up the rate of certain social processes. In the remainder of this section, we discuss the implications of these mechanisms and outline how they come in competition as connectivity across regions increases.

The first mechanism to explain different paces of social dynamics relies on a varying complexity of interactions between speakers within each Galician region. According to numerical results, underlying complex interactions will relax slower to their equilibria than simpler ones[36]. We propose that the percentage of urban population partly captures this internal complexity and suggests that Fig. 1c is empirical evidence for the results in ref. [36]. Slowing down certain social dynamics can be desirable for policy makers. For example, asymptotic trends might predict the extinction of one tongue, but a long half-life of the shift dynamics affords a chance of survival if conditions change in the future. To leverage this mechanism, the richness of interactions between people must increase as globalization proceeds. Certain technologies hamper the complexification of human interaction—e.g.

television offers a mostly uniforming, one-way channel. Other technologies enhance clustering, thus bridging length and time scales and communication between clusters—as the phone does in urban (yet not rural) areas[41]. The internet, through its echo chamber effect, has also proved a powerful tool at forming clusters across long geographical distances.

We can look at these results from a complementary perspective. We propose that, when tracking social processes on isolated communities, the pace of opinion dynamics could be a proxy of that community's internal complexity. To do this carefully we need several similar communities (as in this paper) to be sure that we compare equivalent dynamical rates. The results in ref. [36] should hold true generically for arbitrary opinion dynamics, not only language shift. In the real world, some trends sweep a region faster than others. We could use fast processes to gain an insight on the underlying social complexity of each region, and thus learn how to influence other, slower ongoing processes in the same places.

The second mechanism to induce different rates in the dynamics of language shift emerges out of a systemic view of Galicia. Within a network, each region becomes a node and speakers feel the influence of all other nodes. Figure 2 shows how a moderate all-to-all connectivity induces dynamics such that urban areas evolve slower than the rural ones, just as in the observed data. It could be thought that connecting all regions would homogenize the network—as it happens in Axelrod's model of cultural dissemination[10]. But a moderate connectivity is capable of introducing heterogeneity in the speed at which different regions evolve—not necessarily on their steady states, though. Extending this to a larger-scale picture, some level of globalization can induce heterogeneity in the rate of certain social dynamics of the system.

To introduce the simplest network structure, we couple every Galician region to a mean field virtual node with a strength $K_i = \alpha(1 + \beta u_i)$, where $\alpha > 0$ provides a baseline connectivity and $\beta u_i$ increases the strength proportionally to a node's urban population. Table 1 summarizes different scenarios depending on $\alpha$ and $\beta$, which govern the connectivity of the model. For both $\alpha \to 0$ vs. $\beta \to 0$, the influence of the network vanishes. For $\beta = 0$, the connection of each node to the large-scale dynamics is not influenced by its urban population. As discussed above, some technologies such as TV or radio can contribute to such egalitarian connectivity. The same could be said about the internet if it

| | $\alpha \ll 1$ | $\alpha \gg 1$ |
|---|---|---|
| **Table 1 Different possible combinations for the parameters $\alpha$ and $\beta$** | | |
| $\beta \ll 1$ | Sparse, egalitarian. Internal dynamics dominate. | Well connected, egalitarian. |
| $\beta \gg 1$ | Sparse, non egalitarian. | Well connected, non-egalitarian. Network dynamics dominate. |

The situation of Galicia could be included in the first line, $\beta \ll 1$ in an intermediate case between the two limiting situations considered

could reach everywhere simultaneously. Unfortunately, such technologies are often deployed faster in urban areas because a smaller cost makes them available to more users—meaning that they can contribute to create non-egalitarian networks. Roads, similarly, usually connect big cities first. If policy makers would want to achieve egalitarian yet connected societies, they would need to counterbalance these natural patterns of technology deployment.

Figure 3e, f are maps, in the $\alpha$ vs. $\beta$ space, that could guide policy makers to achieve desired rates at which social dynamics (notably language shift) unfold. As suggested above, we might desire slower dynamics to increase the chance that threatened tongues survive. Alternatively, we might choose to speed up such processes to end conflicts. A moderate baseline connectivity ($\alpha \neq 0$) could achieve networks with slower urban dynamics. Our empirical data shows that this is the Galician case, suggesting that this sociolinguistic system has an effective, overall low connectivity to network-wide dynamics.

As either $\alpha$ or $\beta$ increase, urban areas tend faster to their steady state than the rural ones. This is indicated by the negative slopes ($b < 0$) in the regression lines $\lambda = a + bu$ (top-right corner, Fig. 3e). In this regime urban nodes are so much more connected that they have short response times to external perturbations. Extremely high connectivity confers more relevance to the diffusion than to each region's internal dynamics, hence the importance of internal complexity dwindles. We do not observe this in our data, suggesting again that influences across Galician regions are less prominent—despite all the globalization dynamics and technology-driven improvements in communication during the last decades. As discussed above, certain technologies connect cities easier and earlier than other regions. Cities are also natural economic engines that boost a series of techno-logical and economic processes[11–13]. As globalizing forces become important, the contextual relevance of cities will rise, hence speeding up their internal dynamics as Fig. 3e predicts. But the very factor that increases a region's contextual relevance (its urban tissue) is the same that would slow down its pace due to the internal complexity mechanism. This points at a competition between organization scales and suggests that a city within broader systemic dynamics is provided with a feedback loop that could fine-tune the rate of its social processes. This fine-tuning would result out of a tension between internal complexity and contextual relevance of the city. More research is necessary to clarify the effects of this interplay and how it might influence the development of cities within a broader context.

We have introduced two mechanisms worth harnessing to speed up or slow down social dynamics. These relate to different organization scales. A competition between these scales looms for certain dynamical configurations. Our discussion stems from language shift dynamics, whose mathematical models (hence phenomenology) also applies to opinion dynamics. As globalization proceeds, competitions between the dynamics emerging from different organization scales should become more prominent. While our study case is geographically located, competitions between scales are also relevant in the corporate world. The possibility of having similar phenomenology affecting business organization or corporations, where structure often bridges across

scales, highlights the relevance of the mechanisms and potential conflicts studied here.

As a useful summary for policy making, our research suggests the following: If we wished to sustain ongoing social dynamics, an effort should be made to enrich the microscopic interaction network (thus harnessing the internal complexity mechanism). If, instead, we wished to accelerate the resolution of such dynamics, the diversity of interactions should be hampered.

## Methods

**Model equations**. In this section, we collect some mathematical details that are needed to follow the main Results of our work. These entail some of our equations, their stability, etc., or other procedures to study the stability in networked systems; they have largely been already established in previous papers.

As advanced in the main text, we build our results upon recent literature that models language shift by means of differential equations[15–29] (see ref. [6] for a review). Early results by Baggs and Freedman[20,21] dealt more prominently with abstract aspects of such mathematical models. For example, they found general conditions for the existence of language coexistence or minority language survival, even for models whose explicit functional dependence was not fully provided. Starting with reasonable yet general assumptions, they show that models must exist in which bilingualism is a stable feature and both languages can survive in the long term. The work of Abrams and Strogatz[22] gave a great push to this mathematical approach to language shift, notably because their parsimonious equations (AS model) was successfully compared to several and varied real-world data sets. The AS model is arguably the simplest, yet meaningful model of language shift that can be written. In its simplicity, it sacrifices the ability to fit bilingual populations. Also, it does not contemplate the possibility of stable coexistence. While other models exist, two interesting ones stem directly from the AS model: The one that we use in this work[15] and an alternative one introduced by Minett and Wang[25]. The Minett and Wang model only admits transitions from the monolingual groups to the bilingual one and vice-versa—never directly between the monolingual groups. In this model, coexistence and stable bilingualism is still ruled out.

The equations that we base our results in do allow transitions between any linguistic groups, including between the monolingual groups. This model has parameter combinations in which both languages can coexist in the long term, always alongside a stable bilingual group—but it is by no means the only model contemplating stable bilingualism, as illustrated by the Baggs and Freedman equations with varied degrees of abstraction[20,21]. We prefer this model, partly, because its stability has been thoroughly studied[16,30,31] and because it has been compared to several real collections of linguistic data, including records from Galicia.

Let us assume the existence of two languages, X and Y, with a fraction of monolingual speakers x and y. Bilingual speakers (B) constitute a fraction $b = 1 - x - y$ so that the total population is normalized. Two differential equations suffice to describe the flux between both monolingual groups and the bilingual one:

$$\begin{aligned} \frac{dx}{dt} &= yP_{YX} + bP_{BX} - x(P_{XY} + P_{XB}), \\ \frac{dy}{dt} &= xP_{XY} + bP_{BY} - y(P_{YX} + P_{YB}). \end{aligned} \quad (3)$$

Similarly to the Abrams–Strogatz model[22], the likelihoods of changing groups ($P_{XY}$, $P_{XB}$, $P_{YX}$, $P_{YB}$, $P_{BX}$, and $P_{BY}$) depend on how many speakers use the opposite language and on a prestige parameter $s \equiv s_x$ (such that $s_x + s_y = 1$, $s \in [0, 1]$). Additionally, an interlinguistic similarity parameter $k \in [0, 1]$ tells us, of all the speakers that learn a new language, the fraction of them that retains the previous one. This parameter $k$, hence controls the access to the bilingual group. (See Supplementary Information for more details.) Following[15], the functions ($P_{XY}$, $P_{XB}$, $P_{YX}$, $P_{YB}$, $P_{BX}$, and $P_{BY}$) are given by

$$\begin{aligned} P_{XB} &= ck(1-s)(1-x)^a, \\ P_{YB} &= cks(1-y)^a, \\ P_{BX} &= P_{YX} = c(1-k)s(1-y)^a, \\ P_{BY} &= P_{XY} = c(1-k)(1-s)(1-x)^a. \end{aligned} \quad (4)$$

Prestige and interlinguistic similarity ($s$ and $k$) are the parameters that affect the stability of the model the most. Together with the initial conditions, they determine whether both languages, or only one (and which), survive. Two more parameters enter this model in Eq. (4): The exponent $a$ (termed volatility[6,28]) conveys an idea

of how prone speakers within a population are to attempt a language change. Finally, a parameter $c$ sets the temporal scale at which the dynamics are resolved. The larger $c$ the faster the dynamics unfold, while a small value implies a slow decay towards the stable state of the system. This parameter does not affect the stability of the model.

**Network**. In continuous media, the diffusive transport obeys Fick's law where the flux is proportional to the concentration gradient:

$$
\begin{aligned}
j_x &= D_x \nabla^2 x(\vec{r}, t), \\
j_y &= D_y \nabla^2 y(\vec{r}, t).
\end{aligned}
\tag{5}
$$

In networks and following ref. [38], the diffusive transport to a certain node $i$ is equal to the sum of every flux between node $i$ and the rest of nodes connected to it for each of the species involved

$$
\begin{aligned}
\sum_{j=1}^{N} A_{ij}\left(x_j - x_i\right) &= \sum_{j=1}^{N} L_{ij} x_j, \\
\sum_{j=1}^{N} A_{ij}\left(y_j - y_i\right) &= \sum_{j=1}^{N} L_{ij} y_j,
\end{aligned}
\tag{6}
$$

which is the equivalent of Fick's Law in networks[38]. The Laplacian matrix $L_{ij}$ of a network is a real, symmetric and negative semidefinite matrix (strictly negative eigenvalues), whose elements are given by $L_{ij} = A_{ij} - d_i \delta_{ij}$ [38] (with $\delta_{ij}$ the Kronecker delta). The adjacency matrix $A_{ij}$ is the most used method for the representation of networks. The presence of links is given by $A_{ij} = 1$ and the absence is given by $A_{ij} = 0$. $d_i = \sum_{j=1}^{N} A_{ij}$. The connectivity or degree of the node $i$ is $d_i = \sum_{j=1}^{N} A_{ij}$ and it measures the number of nodes connected to the node $i$.

In the mean field approximation, used in the calculations throughout the paper, the detailed description of each node in the network is neglected and each node is considered to be coupled to a mean field (obtained as the average of all the individual values for each node). This is the expression used in Eq. 2.

**Linear stability analysis**. To analyze the linear stability of a system in the classical case of a continuous media, perturbations to the steady state are decomposed in a set of Fourier modes which represent plane waves with different wave numbers. In discrete space, the role of the wave modes and of the wave numbers[38] is played by, respectively, the eigenvectors $\phi^{\alpha} = \left(\phi_1^{\alpha}, \dots, \phi_N^{\alpha}\right)$ and the eigenvalues $\Lambda_{\alpha}$, $\alpha = 1, \dots, N$, of the Laplacian matrix $L_{ij}$.

The eigenvalues $\Lambda_{\alpha}$, $\alpha = 1, \dots, N$, and the associated eigenvectors $\phi^{\alpha} = \left(\phi_1^{\alpha}, \dots, \phi_N^{\alpha}\right)$ of the Laplacian matrix $L_{ij}$ are determined by:

$$
\sum_{j=1}^{N} L_{ij} \phi_j^{\alpha} = \Lambda_{\alpha} \phi_i^{\alpha}
\tag{7}
$$

All eigenvalues are real and non-positive.

In the mean field approximation, the diffusive flux is expressed by $\sum_{j=1}^{N}(\acute{x} - x_i)$ where $\acute{x} = \frac{1}{N}\sum_{j=1}^{N} x_j$ is the mean value of the x-variable. The elements of the adjacency matrix can be written as $A_{ij} = 1/N$ if $i \neq j$ and $A_{ij} = 0$ if $i = j$. The degree is $d_i = (N-1)/N$. Thus, we can write $L_{ij} = 1/N$ if $i \neq j$ and $L_{ij} = (1-N)/N$ if $i = j$. Established this network, all eigenvalues are $-1$.

As in continuous media, we analyze the linear stability[37,38] perturbing the uniform state $(x|i, y_i) = (x_0, y_0) + (\delta x_i, \delta y_i)$ and substitute this into the reactive-diffusive equations:

$$
\begin{aligned}
\frac{\partial}{\partial t}(\delta x|i) &= f_x \delta x_i + f_y \delta y_i + K_i \sum_{j=1}^{N} L_{ij} \delta x_j, \\
\frac{\partial}{\partial t}(\delta y|i) &= g_x \delta x_i + g_y \delta y_i + K_i \sum_{j=1}^{N} L_{ij} \delta y_j,
\end{aligned}
\tag{8}
$$

where the subindexes $x$ and $y$ of the functions $f$ and $g$ are the coordinates with respect to which we differentiate: $f_x = \partial f / \partial x$ and so on. Perturbations are expanded over the set of $\alpha$ Laplacian eigenvectors like plane waves

$$
\begin{aligned}
\delta x_i &= C_{\alpha} e^{\lambda_{\alpha} t} \phi_i^{\alpha}, \\
\delta y_i &= C_{\alpha} B_{\alpha} e^{\lambda_{\alpha} t} \phi_i^{\alpha}.
\end{aligned}
\tag{9}
$$

Introducing these definitions into the system of Eq. (8) above:

$$
\begin{aligned}
\lambda_{\alpha} C_{\alpha} e^{\lambda_{\alpha} t} \phi_i^{\alpha} &= f_x C_{\alpha} e^{\lambda_{\alpha} t} \phi_i^{\alpha} + f_y C_{\alpha} B_{\alpha} e^{\lambda_{\alpha} t} \phi_i^{\alpha} + K_i \sum_{j=1}^{N} L_{ij} C_{\alpha} e^{\lambda_{\alpha} t} \phi_j^{\alpha}, \\
\lambda_{\alpha} C_{\alpha} B_{\alpha} e^{\lambda_{\alpha} t} \phi_i^{\alpha} &= g_x C_{\alpha} e^{\lambda_{\alpha} t} \phi_i^{\alpha} + g_y C_{\alpha} B_{\alpha} e^{\lambda_{\alpha} t} \phi_i^{\alpha} + K_i \sum_{j=1}^{N} L_{ij} C_{\alpha} B_{\alpha} e^{\lambda_{\alpha} t} \phi_j^{\alpha}.
\end{aligned}
\tag{10}
$$

Dividing by $C_{\alpha} e^{\lambda_{\alpha} t}$ and using the definition of eigenvalue, Eq. (7), the result is $N$ independent linear eigenvalue equations for the different $\alpha$ normal modes that, in

matrix notation look like:

$$
\lambda_{\alpha} \begin{pmatrix} 1 \\ B_{\alpha} \end{pmatrix} = \begin{pmatrix} f_x + K_i \Lambda_{\alpha} & f_y \\ g_x & g_y + K_i \Lambda_{\alpha} \end{pmatrix} \begin{pmatrix} 1 \\ B_{\alpha} \end{pmatrix}
\tag{11}
$$

The information about the evolution of the diffusive system can be found analyzing the matrix $J$

$$
J = \begin{pmatrix} f_x + K_i \Lambda_{\alpha} & f_y \\ g_x & g_y + K_i \Lambda_{\alpha} \end{pmatrix}
\tag{12}
$$

The growth factor $\lambda_{\alpha}$, $\alpha = 1, \dots, N$, is given by the eigenvalues of the matrix $J$. The growth factor, by construction, measures the rapidity of evolution to a new stationary state after a perturbation. Note that the Laplacian eigenvalue $\Lambda_{\alpha}$ multiplies $K_i$. If the mean field network approximation is considered (i.e., $\Lambda_{\alpha} = -1$), we can, thus, rewrite the matrix $J$ as

$$
J = \begin{pmatrix} f_x - K_i & f_y \\ g_x & g_y - K_i \end{pmatrix}
\tag{13}
$$

From the characteristic polynomial of the matrix $J$, $det(J - \lambda_{\alpha}I) = 0$, $(f_x - K_i - \lambda_{\alpha})(g_y - K_i - \lambda_{\alpha}) - f_y g_x = 0$, we obtain the expression for the eigenvalues of the matrix $J$, $\lambda_{\alpha}$, or growth factors

$$
\lambda_{\alpha} = \frac{1}{2}\left[ f_x + g_y - 2K_i \pm \sqrt{4 f_y g_x - \left(f_x - g_y\right)^2} \right]
\tag{14}
$$

This expression allows to calculate the growth factors directly from the model equations and parameters, it gives a direct measure of the rhythm of evolution to a new stationary state after perturbing the system. In conclusion, linearizing the system around the fixed point $(x_0, y_0)$, the growth factor, $\lambda_{\alpha}$, is calculated as the eigenvalue of the matrix $J$. Consequently, the growth factor $\lambda_{\alpha}$ depends on the degree of globalization $K_i$.

## Data Availability
Authors can confirm that all relevant data are included in the paper and its Supplementary information files. In any case, the data are also available upon request from the authors.

## Code availability
The code is available from the authors on request.

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

## Acknowledgements

We gratefully acknowledge financial support by the Spanish Ministerio de Economía y Competitividad and European Regional Development Fund under contract MAT2015-71119-R AEI/FEDER, UE, and by Xunta de Galicia under Research Grant No. GPC2015/014. A.P.M. and M.M.J. are part of the CRETUS Strategic Partnership (AGRUP2015/02) and J.M. is part of the AeMAT Strategic Partnership (Grant No. ED431E2018/08), both supported by Xunta de Galicia. All these programs are co-funded by FEDER (UE). We also thank M. C. Parafita Couto, from the Leiden University Centre for Linguistics, for her help. J. M. acknowledges support from the Xunta de Galicia under the Strategic Grouping AEMAT (Grant No. ED431E2018/08).

## Author contributions

M.M.J. and L.F.S. contributed equally to this work. All authors wrote the paper and contributed to the production of the figures. M.M. and A.P.M. developed the part of the work that concerns the mean-field network models. L.F.S. and J.M. developed the part of the work that concerns the analysis based on the internal complexity of the different regions.
