## [Transparent Peer Review File · Nature Communications]

Reviewers' comments:

Reviewer #1 (Remarks to the Author):

The main claim of the paper is that sociolinguistic phenomena such as language shift can often involve interactions across different scales and as such result in social and linguistic changes that can be tracked over time. The paper focuses specifically on dynamics of how language shift takes place and it uses historical data from Galician and Spanish speakers in Galicia (a bilingual community in northwest Spain), to study the rate at which shift dynamics take place. The main conclusions seem to be that language shift correlates inversely with the internal complexity of a region which is linked largely to the proportion of urban areas in the region. The key finding is that the more complex the areas, the more likely it is to sustain a heterogeneous dynamic over time. Conversely, less complex areas converge faster. The authors further explore this model by introducing an additional area of complexity, namely, geographic migration. They argue that this introduces "a conflict between the internal complexity of a region and its contextual relevance in the migration network. Harnessing these sociodynamical features will prove useful in policy making to limit conflicts". It is not fully clear what "conflict" the authors are referring to and how in practical terms the findings can be of relevance to policy making in Galicia.

The paper is novel and the analysis of the language shift in Galicia using the proposed methods outlined in the paper has the potential to complement existing work in sociolinguistics in particular about the dynamics of language survival and loss in the Galician contexts. It would be good if the authors provided some background to that literature so as to present the current state of the field and how their approach can add to that. It would be useful if the authors could provide more detail on how their study differs to others that already exist and how their study can provide new insights into understanding the dynamics of language shift in Galicia and perhaps how this model could be applied to studies on language shift more broadly as it relates to other bilingual communities where there is a minoritized language.

More discussion would also be welcome on terms like globalisation and scale. In sociolinguistics there has been a lot of work done on this e.g. Blommaert 2010. Similarly, the terms sociolinguistics does not explain and neither does language shift. The latter is often problematised now. The authors would also need to show awareness of the literature in language maintenance and shift, tracing it from the classical studies of Fishman (1991) up to more contemporary debates which question his Reversing Language Shift model.

Reviewer #2 (Remarks to the Author):

Please see attached document for full review. In short, the technical results and interpretation are worthy of publication, but the prose and organization will require revision in order to clearly communicate ideas. As it stands, it is difficult to parse, especially on first read-through. After several readings I have a more thorough understanding of the work and believe it is a significant advancement in the field. I was intrigued by your results and very much enjoyed learning about your work.

Reviewer #3 (Remarks to the Author):

The submitted paper has the general goal of contributing to the understanding (and harnessing) of social tensions, through the study of the mechanisms underlying language shifts, with the help of some language dynamics models. This and the disappearance of languages and cultural heterogeneity are currently topics of high interest.

However, I find the paper inconsistent in various points - especially in the second model considered - and not suitable for publication.

In greater detail, the paper studies two different models which are to be considered separately.

First Model. It features the role of internal complexity on language shift, on the base of the particular model introduced in Ref. [15]. The findings, concerning unexpected correlations between the pace of language shift and the rural/urban character of the locations, are interesting and are presented clearly enough. But they also deserve further attention and studies (and probably a more detailed discussion in a paper) and are based on one particular model (among various one proposed in the literature).

Second Model. It should describe (the additional effect of) individuals moving between towns/regions, depending on what is referred to as the "contextual relevance" K_i . However, it seems to me that this model is not a simple diffusion model and for this reason it just does not fit the paper, representing something different from what the authors claim.

First, Equations (1) do not conserve the total population sizes and therefore do not describe only diffusion but also some underlying (not specified) populations dynamics. Migrations should conserve separately both x and y total populations, but neither x nor y - nor the total population - are conserved. To conserve population, one should not use coefficients K_i in Eqs. (1) but symmetrical coefficients k_{ij} (of course inside the sum over j).

A dependence of k_{ij} on both i and j would also reflect an actual contextual relevance of e.g. a city. If the diffusion flux between two towns (i) and (j) is simply proportional to the difference ($x_i - x_j$), this means really a free diffusion and therefore a contextual equivalence between the towns (a coefficient K_j and a coefficient K_j would just weight different but otherwise free diffusion processes). I do not comment on the specific results obtained from the second model, since I do not know what they really represent.

In conclusion, I see this paper as made up of two distinct contributions.

A first part (the first model) containing valuable results. If additional results and explanations would be provided on possible interpretations of the findings, this part may be further elaborated into a short but interesting article.

A second part, which does not seem to be consistent with the rest of the paper, which in case would need drastic revisions already at the level of model definition.

Review of “Internal complexity versus globalization as a coarse graining of social dynamics”

General comments

The authors present an analysis of language shift in Galicia using two related models which describe dynamics within the region. In the first model, district dynamics evolve independent of one another, and time scales determined by fitting to data provide information about each district. In the second, districts are coupled to one another, with coupling strengths based on the urban/rural characteristics of each district. The specific case of Galicia is examined by fitting parameters, then the general model behavior is analyzed.

The results introduced are an interesting and valuable addition to the field. Both the theoretical methods used and the results specific to Galicia are noteworthy. The work is original and novel, extending previous work by the authors and others, in particular references 15–28 as cited in the paper.

As submitted, the paper requires significant revision in terms of writing and organization. Some problems are with grammar and vocabulary, which I assume is due to language. Perhaps these can be remedied in the copyediting process, but they currently distract from the content of the paper, especially when word choice makes meaning ambiguous. Other problems relate more to the content. In particular, the background and interpretation can generally be condensed, and the technical details should be more carefully introduced so that an uninitiated reader can follow the ideas. For example, equations are referenced long before they are given and variables are used before they are defined. Some of the theoretical foundation is given in the Results section, while some is given in Methods. Suggested revisions ought to reduce the length of the paper, which as it stands is overly long. Detailed edits are given below. Generally there is significant room for the prose to be made both more concise and more clear.

If revisions are satisfactory I expect that this work would be acceptable for publication, based on its mathematical/sociological content.

Specific comments/questions/suggestions

Please see the annotated pdf for instances of unclear wording/grammar, which I have highlighted throughout. One common problem is misuse of adjectives vs. adverbs. Another is the use of the word “shall” throughout the paper, which is anachronistic/archaic. Replace with “may,” “must,” “should,” “will,” etc. depending on context.

Abstract

- Unclear what you mean by “more heterogeneous,” without having first read the rest of the paper.
- Claiming this “will prove useful” is a bit strong, I’d say “may prove useful.”

1st section

- Paragraph 3: Which model “hints at...,” yours or Axelrod’s?
- Paragraphs 1–3 could be condensed.
- Paragraphs 4 and 5 are nearly identical, this must have been a mistake (omit paragraph 5).

Results

- Paragraph 1: Use of “XX century” instead of “20th century” or “twentieth century” is nonstandard and therefore disorienting. (It seems the Roman numerals are more typical in Spanish.)
- Paragraph 3: Introducing the variables and parameters at the end of the paragraph is unnecessary. They don’t have meaning to the reader (yet).
- Paragraph 4:
 - In the second sentence, are you now describing your own work? Make this clear by rewording to avoid passive voice.
 - The index i is explained late; reword to explain that parameters are for an individual region at start of paragraph.
 - You introduce groups A, B, C, and D in this paragraph using qualitative descriptions. I expected more precise, quantitative definitions at some point, did I miss this? Does “mostly bilingual, with Galician preponderance” simply mean that $b > 0.5$ and the fraction speaking Galician is greater than that speaking Spanish?
- Paragraph 5: The final sentence is unclear.

Effect of internal complexity

- Paragraph 2:
 - Explanation of what the dots mean in figure 1b is unclear. Are these locations of particular cities/towns, or general well-known areas, or are you marking the general region? Can you be more specific?
 - Is 5000 inhabitants an accepted definition of “urban,” or was this chosen arbitrarily (or for another reason)?
- Paragraph 3: Referencing equations that have not been given (and won’t be until nearly the end of the paper) makes this frustrating to read.

I recommend moving the first part of your Methods section, which gives these equations and explains them, to this section. In fact, it may be possible to eliminate the Methods section altogether, moving all of the content to Results.

Alternately, you could explain the ideas behind the equations without referring to the equation numbers (and thus the specific equations) so extensively (perhaps referring to the model using a descriptive term or name), and save the technical discussion for Methods. If you do this, I would recommend moving your upcoming technical discussion of equation (1) to Methods as well. The current organization feels disjointed to an uninitiated reader.

Another possibility (I’m guessing here) is that the current organization is because the details in Methods are not novel, and those in Results are novel. However, this is not clear to the reader (both sections seem to be treated similarly, the novelty is not explained).

Migration network and contextual relevance

- Paragraph 1:
 - The variables x and y have not yet been introduced, so these equations are impossible to understand without jumping ahead to the Methods section. See previous comments on organization.

- Explanation of K_i is unclear due to word choice. It seems to me that this parameter measures the relative effects of internal and external dynamics. Large K_i indicates stronger coupling to other regions, while small K_i indicates weak coupling, meaning that dynamics are largely driven by the model in eqns. 2-3 alone, where the districts are treated independently.
- Paragraph 2: I don't understand the meaning of the second sentence: doesn't K_i effect how likely speakers are to diffuse? Are you saying that K_i is the same for every district i ?
- Paragraph 3: By "prevalent," do you mean "dominant?"
- Paragraph 5:
 - Your choice for how K_i depends on u_i is not sufficiently explained. The post-hoc explanation of what each parameter represents is fine, but I don't see clearly where this came from and why it's better than other potential options. Needs more justification.
 - Where did the specific values for α and β come from? Are these from a fit to data, or chosen arbitrarily? Unclear.
 - Where do you "show the average?" The average of what variable(s)?

Competing mechanisms

- Paragraph 2:
 - Explain what you mean by "another case." Is this just a different choice of values for α and β ?
 - Does p represent the slope of your line of best fit? If so, state this explicitly.
 - You refer to plots by their horizontal/vertical axis variables, which is reasonable but difficult to follow due to formatting. Currently it reads like the subtraction of the two variables. Perhaps this could be fixed by using a hyphen rather than a minus sign/en-dash, or by rewording like " x versus y ." This comes up again in the following paragraphs/sections.
 - Using "plane" to describe a color map/contour map/surface is confusing.

Discussion and Conclusions

- Paragraph 3: Last sentence unclear, perhaps unnecessary.
- Paragraph 4: First and third sentences unclear.
- Paragraph 6: Case of $\beta \rightarrow 0$ needs clarification. The following sentences should be reworked as well. In general, the speculation and value judgements here seem unnecessary and not directly connected to your results.
- Paragraph 7: Last sentence is unclear, where does this come from?

Methods: Model equations

- See previous comment on organization.
- Sentence before eqns. 3 is unclear, what do you mean by "controls access?" How this sentence transitions into the equations is difficult to parse.
- Equations given in (3) are similar to those in another paper, correct? Why not cite this source more explicitly here, or explain in more detail? Something like "In order to model..., we follow the methods from..., where..."

Methods: Network

- Paragraph 2: You reuse k_i here, should be careful to note this (or change notation).
- Why not use the more conventional term “degree” to denote the number of connections of a node rather than “connectivity?”
- Your definition of the graph Laplacian differs from convention: Wolfram gives the definition $L = D - A$, which has sign opposite yours (see <http://mathworld.wolfram.com/LaplacianMatrix.html>). Why is this?

Methods: Linear Stability Analysis

- Some of the paragraph breaks here seem to be unintentional (after equations).
- Does \bar{x}_j actually depend on j ? (typo?)
- After eqn. 7, as written this is not a sentence.
- Before eqn. 11, what do you mean by “J matrix?” Is this a conventional technical term (as in a Jacobian matrix), or do you mean something more arbitrary like “the matrix J?”
- The resulting eigenvalue in eqn. 13 should be explained/interpreted. I’m assuming it is meaningful because you included it, so what is important in this formula? What information does this give?

Author Contributions

- Details are only given for two of the four authors.

Figures

- Figure 1b: See previous comment about dots and meaning of “prominent.”
- Figure 1c: Do error bars show 2 standard deviations from the mean?
- Figure 2a is missing when it is meant to be given together with 2b and the caption, though it did show up with the separately attached figures. Perhaps this is a technical glitch.

Supplementary Information

- Need more description of what is given in the table. Clearly it is quantitative information, but spell out in a few sentences whether these are best fits, for which equations, etc.
- The caption seems unnecessarily complicated. Why not include this information (district names) in the table itself? You could make a landscape-oriented table if necessary.
- The end of the table is missing a reference.

REVIEWER #1:

Reviewer #1: (Remarks to the Author):

The main claim of the paper is that sociolinguistic phenomena such as language shift can often involve interactions across different scales and as such result in social and linguistic changes that can be tracked over time. The paper focuses specifically on dynamics of how language shift takes place and it uses historical data from Galician and Spanish speakers in Galicia (a bilingual community in northwest Spain), to study the rate at which shift dynamics take place. The main conclusions seem to be that language shift correlates inversely with the internal complexity of a region which is linked largely to the proportion of urban areas in the region. The key finding is that the more complex the areas, the more likely it is to sustain a heterogeneous dynamic over time. Conversely, less complex areas converge faster. The authors further explore this model by introducing an additional area of complexity, namely, geographic migration. They argue that this introduces "a conflict between the internal complexity of a region and its contextual relevance in the migration network. Harnessing these sociodynamical features will prove useful in policy making to limit conflicts". It is not fully clear what "conflict" the authors are referring to and how in practical terms the findings can be of relevance to policy making in Galicia.

Answer: First of all, we thank this referee for having pointed out several useful ideas, that helped us to better sustain our work. We have introduced his/her suggestions as much as possible to fit them with the requests of the other two referees.

We answer following his/her requests:

Referee 1: It is not fully clear what "conflict" the authors are referring to

Answer: The referee is right, we were using the same word ("conflict") for different concepts: a more mathematical one (the competition among the different dynamics of networks) and the social one (between communities of speakers).

We are using therefore the word "competition" for the relationship between the internal complexity and the global network, and the word "conflict" to address the relationship between communities of speakers.

Referee 1: and how in practical terms the findings can be of relevance to policy making in Galicia.

Answer: This work intends to describe in a quantitative and general way the dynamics of language shift. In the specific case of Castilian-Galician, the results are of interest for the office of the Secretary General of Linguistic Policy of Galicia, who, in fact, is aware of our work and has particular interest in the interrelation of these languages depending on the urban/rural framework, in order to design specific campaigns to promote Galician tongue in places under more risk of loss of this language.

We believe our results can be of relevance as they describe how complexity plays a role in the evolution of languages and, in general, conflicts. There are several ways to modify the complexity both at a local level as well as in the network context; i.e. access to internet, roads and transport facilities, etc. This understanding can be used as a guideline to attenuate conflicts or protect minority endangered languages. We include now a tip for policy makers in the discussion section, about counterbalancing the natural pattern of how technology is implanted.

Referee 1: The paper is novel and the analysis of the language shift in Galicia using the proposed methods outlined in the paper has the potential to complement existing work in sociolinguistics in

particular about the dynamics of language survival and loss in the Galician contexts. it would be good if the authors provided some background to that literature so as to present the current state of the field

Answer: There is literature on the linguistic aspects of Galicia, which ranges from aspects related with regional identity, conflicting views about the value associated with Galicia language, the linguistic history of Galicia and, of course, on data use. We are now including references [17] and [18]).

Referee 1: (...) and how their approach can add to that. It would be useful if the authors could provide more detail on how their study differs to others that already exist and how their study can provide new insights into understanding the dynamics of language shift in Galicia and perhaps how this model could be applied to studies on language shift more broadly as it relates to other bilingual communities where there is a minoritized language.

Answer: Our approach comes from a different angle, compared to the available ones about Galicia: it starts from basic considerations that leads to a mathematical model, to be contrasted with historical data of the number of speakers of three different groups: Galician and Castillian monolinguals and bilinguals. This is the first approach of this kind done in Galicia. Given that the model uses datasets of speakers of these three groups, it can be applied to language shift studies everywhere where there is a minoritized language. We have tried to make this now clearer in the text.

Referee 1: More discussion would also be welcome on terms like globalisation and scale. In sociolinguistics there has been a lot of work done on this e.g. Blommaert 2010. Similarly, the terms sociolinguistics is not explains and neither is language shift. The latter is often problematised now.

Answer: Certainly sociolinguistics (the study of the influence that society has on the way language is used and the society's effect on language) is blooming, in order to incorporate the influence of globalization. Blommaert, in fact, has launched the idea that stability in social, cultural and linguistic formations can no longer be presupposed because of the disappearance of predictability.

The book of Blommaert, mentioned by the referee, is sure a key work (that we are now citing). As Bloommaert says, the world has not become a village, but rather a complex web of villages, towns and different types of settlements, connected by material and symbolic ties, whose effects are not known; a complexity that needs to be examined and understood. The aim of uncovering such mechanisms is what guides our work. For this purpose we use language shift in our basic model, considering changes between the three groups (2 monolinguals and 1 bilingual) that are described in the historical data.

This need for theoretical approaches and new research methods is perceived also from other works (we include now reference [39]), which state that classical sociolinguistic ill-fits non-western cases but also state that it is difficult to apply to large urban areas in the western world today. The urban/rural division within a state actually pertains to a more fundamental distinction, namely that between two worlds. We share this awareness of a western dominance in sociolinguistic theory-making and the need for a better understanding of this urban/rural division. For this reason, we think that our proposal of a mathematical tool could be useful, as it is focusing on this relationship between urban and rural areas. These reflections are now included in the discussion.

Referee 1: The authors would also need to show awareness of the literature in language maintenance and shift, tracing it from the classical studies of Fishman (1991) up to more contemporary debates which question his Reversing Language Shift model.

Answer: We appreciate this observation. We are now referring in the introduction to the Reversing Language Shift model of Fishman and his influential Graded Intergenerational Disruption Scale (GIDS), which has undergone several adaptations. We have detected that there may be some criticism of such model, on considerations about if changes are gradual, mechanical, evolutionary or cumulative. Also, to the possible danger that the neatness of the GIDS leads to the illusion that reversal is easy, with simple solutions and easy decisions; but we have not found specific citations, even after some consults with a researcher from the Leiden University Centre for Linguistics. We have therefore opted for the addition of reference [33], which gives an overview about modeling of language shift.

Inspired by this comment of the referee, we are now adding that our mathematical model yields a phase space where, depending on the parameters calculated from the fits to empirical data, we can measure quantitatively the degree of risk of the weak language.

REVIEWER #2:

Answer: All comments marked in a PDF version of the previous manuscript by Reviewer 2 have been included in this new version.

Reviewer #2: General comments:

The authors present an analysis of language shift in Galicia using two related models which describe dynamics within the region. In the first model, district dynamics evolve independent of one another, and time scales determined by fitting to data provide information about each district. In the second, districts are coupled to one another, with coupling strengths based on the urban/rural characteristics of each district. The specific case of Galicia is examined by fitting parameters, then the general model behavior is analyzed.

The results introduced are an interesting and valuable addition to the field. Both the theoretical methods used and the results specific to Galicia are noteworthy. The work is original and novel, extending previous work by the authors and others, in particular references 15–28 as cited in the paper.

As submitted, the paper requires significant revision in terms of writing and organization. Some problems are with grammar and vocabulary, which I assume is due to language. Perhaps these can be remedied in the copyediting process, but they currently distract from the content of the paper, especially when word choice makes meaning ambiguous. Other problems relate more to the content. In particular, the background and interpretation can generally be condensed, and the technical details should be more carefully introduced so that an uninitiated reader can follow the ideas. For example, equations are referenced long before they are given and variables are used before they are defined. Some of the theoretical foundation is given in the Results section, while some is given in Methods. Suggested revisions ought to reduce the length of the paper, which as it stands is overly long. Detailed edits are given below. Generally there is significant room for the prose to be made both more concise and more clear.

If revisions are satisfactory I expect that this work would be acceptable for publication, based on its mathematical/sociological content.

Specific comments/questions/suggestions :

Please see the annotated pdf for instances of unclear wording/grammar, which I have highlighted throughout. One common problem is misuse of adjectives vs. adverbs. Another is the use of the word “shall” throughout the paper, which is anachronistic/archaic. Replace with “may,” “must,” “should,” “will,” etc. depending on context.

Answer: We checked these issues and believe the new version is more accurate.

Reviewer #2: Abstract: Unclear what you mean by “more heterogeneous,” without having first read the rest of the paper.

Answer: Heterogeneous was a poor word choice. We changed to “transitory dynamics”, closer to what we really intended to say.

Reviewer #2: Abstract: Claiming this “will prove useful” is a bit strong, I’d say “may prove useful.”

Answer: Agreed – “may prove useful” has been adopted.

Reviewer #2: 1st section: Paragraph 3: Which model “hints at...,” yours or Axelrod’s?

Answer: Axelrod's. This has been clarified.

Reviewer #2: 1st section: Paragraphs 1–3 could be condensed.

Answer: Some effort was done throughout the manuscript to condense the text but due to the numerous comments from the referees this was not always possible.

Reviewer #2: 1st section: Paragraphs 4 and 5 are nearly identical, this must have been a mistake (omit paragraph 5).

Answer: It was. This has been corrected now.

Reviewer #2: Results: Paragraph 1: Use of “XX century” instead of “20th century” or “twentieth century” is nonstandard and therefore disorienting. (It seems the Roman numerals are more typical in Spanish.)

Answer: All instances of “XX century” have been changed to “20th century”.

Reviewer #2: Results: Paragraph 3: Introducing the variables and parameters at the end of the paragraph is unnecessary. They don’t have meaning to the reader (yet).

Answer: 1. Throughout her or his review, the Referee raises several times issues about the structure of our paper, specifically about the Methods section and the need to refer to it repeatedly from the main text. At some point, the Referee suggests that we eliminate the Methods section altogether, or that we incorporate large portions of it into the Results section.

1. We believe that, in confining most equations to the Methods section, we are following the standard style guidelines of Nature Communications papers. Also, note that ours is a long paper: shifting as much material as possible to Methods allows us to explore with more depth our findings. Because of all these reasons, we have decided to retain the Methods section and to keep referring the reader to it.

2. However, we agree with the Referee that some variables, parameters, or equations were abruptly introduced in our manuscript (e.g. the ones that she or he mentions in the third paragraph of the Results section). We also think that calls to the Methods section were not as timely as necessary throughout the text. We have done a poor job in these two regards. To fix this, we have taken the following actions:
 1. When variables appear for the first time in the main text, brief explanations have been introduced. This makes the text slightly redundant with the Methods section, but we believe that it leads to a better article. We also believe that a detailed explanation of the models, their variables, and parameters is necessary; but that these should not occupy a prominent space in the paper because the relevant findings do not rely heavily on the models used. Hence our insistence on keeping the Methods section.
 2. Throughout the text, we refer the reader to the Methods section more often so that she or he is sure that all the necessary details are provided.

Reviewer #2: Results: Paragraph 4:

Reviewer #2: In the second sentence, are you now describing your own work? Make this clear by rewording to avoid passive voice.

Answer: Fixed.

Reviewer #2: The index i is explained late; reword to explain that parameters are for an individual region at start of paragraph.

Answer: Done.

Reviewer #2: You introduce groups A, B, C, and D in this paragraph using qualitative descriptions. I expected more precise, quantitative definitions at some point, did I miss this? Does “mostly bilingual, with Galician preponderance” simply mean that $b > 0.5$ and the fraction speaking Galician is greater than that speaking Spanish?

Answer: A quantitative definition of groups A-D is now provided.

Reviewer #2: Results: Paragraph 5: The final sentence is unclear.

Answer: Fixed.

Reviewer #2: Effect of internal complexity: Paragraph 2:

Reviewer #2: Explanation of what the dots mean in figure 1b is unclear. Are these locations of particular cities/towns, or general well-known areas, or are you marking the general region? Can you be more specific?

Answer: We are marking the general region. The figure has been modified to make this more explicit. The text now explicitly mentions the rural and urban nature of these areas at large.

Reviewer #2: Is 5000 inhabitants an accepted definition of “urban,” or was this chosen arbitrarily (or for another reason)?

Answer: 5000 inhabitants was chosen based on existing regulation. Locations with 5000 inhabitants or more, new regulations regarding construction and council representation apply. Also, direct inspection of the raw data (see figure below) shows a discontinuity in the series at around 5000 inhabitants (probably due to the actual legislation)

Reviewer #2: Effect of internal complexity: Paragraph 3: Referencing equations that have not been given (and won't be until nearly the end of the paper) makes this frustrating to read.

I recommend moving the first part of your Methods section, which gives these equations and explains them, to this section. In fact, it may be possible to eliminate the Methods section altogether, moving all of the content to Results.

Alternately, you could explain the ideas behind the equations without referring to the equation numbers (and thus the specific equations) so extensively (perhaps referring to the model using a descriptive term or name), and save the technical discussion for Methods. If you do this, I would recommend moving your upcoming technical discussion of equation (1) to Methods as well. The current organization feels disjointed to an uninitiated reader.

Another possibility (I'm guessing here) is that the current organization is because the details in Methods are not novel, and those in Results are novel. However, this is not clear to the reader (both sections seem to be treated similarly, the novelty is not explained).

Answer: In leaving most mathematical details we were complying with the guidelines of Nature Communications. However, the Referee is right and the text did not have enough flow because of the location of the Methods section. We opted for a compromise: the final form of the model equations is now introduced in the main text (in the Results section). Further details are discussed in the Methods. References to other papers that include similar derivations of the model are included throughout.

Reviewer #2: Migration network and contextual relevance: Paragraph 1: The variables x and y have not yet been introduced, so these equations are impossible to understand without jumping ahead to the Methods section. See previous comments on organization.

Answer: See previous replies about what to do with the methods. Nevertheless, the text has been modified to include a brief explanation of the variables were needed.

Reviewer #2: Migration network and contextual relevance: Paragraph 1: Explanation of K_i is unclear due to word choice. It seems to me that this parameter measures the relative effects of internal and external dynamics. Large K_i indicates stronger coupling to other regions, while small K_i indicates weak coupling, meaning that dynamics are largely driven by the model in eqns. 2-3 alone, where the districts are treated independently.

Answer: She or he understood perfectly. This is exactly what this parameter measure. We believe this is clearer in the new version of the manuscript.

Reviewer #2: Migration network and contextual relevance: Paragraph 2: I don't understand the meaning of the second sentence: doesn't K_i effect how likely speakers are to diffuse? Are you saying that K_i is the same for every district i ?

Answer: The model does not describe actual flows of population. The variables in the model (x_i , y_i and b_i) are percentages of population in each node speaking each one of the language options. The term taking into account the effect of the network just describes the influence of other nodes on the distribution of speakers in the original node. I.e. a node that feels from the network a large percentage of speakers of a single language is more prone to move the spectrum of its population in the same direction (but this does not mean actual displacement of the individuals).

It was our fault that this was not clearly explained in the text and we apologize. Now we introduced the appropriate changes in the text.

On the other hand, the values of K_i are region dependent and provide information on how relevant is this node in the context of the whole network. We made them proportional to the percentage of urban population in region i following observations from the previous section. Nevertheless and following referee 3, we tried some different values for K_i and the results observed can be understood with the ideas explained in this manuscript although they do not seem to be so realistic describing the experimental situation. We included these results in the Supplementary Information.

Reviewer #2: Migration network and contextual relevance: Paragraph 3: By "prevalent," do you mean "dominant?"

Answer: Yes. Fixed!

Reviewer #2: Migration network and contextual relevance: Paragraph 5: Your choice for how K_i depends on u_i is not sufficiently explained. The post-hoc explanation of what each parameter represents is fine, but I don't see clearly where this came from and why it's better than other potential options. Needs more justification.

Answer: We agree with the referee that this was not clearly explained in the text. In brief, Fig. 1c shows the direct correlation between the parameter c in the model and the percentage of urban population in each region. Thus, we considered u like a measurement of the complexity of each region as urban populations are more likely to interact with social media or receive inputs from distant regions. The simplest way to introduce this effect in the equations is through the K_i parameter, thus those regions (nodes) with more urban population are more relevant in the network. We introduce the appropriate corrections in the text.

Nevertheless, some other situations could be analyzed. In the Supplementary Material, and upon request of referee 3, we included some new material considering different expressions for the network coupling coefficients and we observe equivalent results. The conclusion here is that the urban population (with the significance of capacity for complexity) is critical to determine the relaxation times to the steady states.

Reviewer #2: Migration network and contextual relevance: Paragraph 5: Where did the specific values for α and β come from? Are these from a fit to data, or chosen arbitrarily? Unclear.

Answer: The actual values of α and β were chosen such that they qualitatively describe the Galician situation. The actual meaning of these parameters becomes clearer in the following section. We added some comments here and reorganize the paragraph to clarify this point.

Reviewer #2: Migration network and contextual relevance: Paragraph 5: Where do you “show the average?” The average of what variable(s)?

Answer: To simplify the visualization of the results, we show the average growth factor (λ_i) across Galician regions included in each one of the four groups (A to D introduced above) versus the average u_i in each group. In the new version of this paragraph, this is clearer explained.

Reviewer #2: Competing mechanisms: Paragraph 2: Explain what you mean by “another case.” Is this just a different choice of values for α and β ?

Answer: Yes, the referee is right. It was a different choice of (α , β) values, nevertheless and following the referee advice, we changed that figure and put instead the corresponding for the same values of (α , β) as in Fig. 2.

Reviewer #2: Competing mechanisms: Paragraph 2: Does p represent the slope of your line of best fit? If so, state this explicitly.

Answer: Yes, it is the slope of the best fit. In the new version of the manuscript, the slope of the best fit is now called b ($\lambda = a + bu$). This was better explained in the text.

Reviewer #2: Competing mechanisms: Paragraph 2: You refer to plots by their horizontal/vertical axis variables, which is reasonable but difficult to follow due to formatting. Currently it reads like the subtraction of the two variables. Perhaps this could be fixed by using a hyphen rather than a minus sign/en-dash, or by rewording like “x versus y.” This comes up again in the following paragraphs/sections.

Answer: We accept the suggestion and modified the text accordingly.

Reviewer #2: Competing mechanisms: Paragraph 2: Using “plane” to describe a color map/contour map/surface is confusing.

Answer: Fixed. Changed “plane” for “space”.

Reviewer #2: Discussion and Conclusions: Paragraph 3: Last sentence unclear, perhaps unnecessary.

Answer: Fixed. Sentence removed.

Reviewer #2: Discussion and Conclusions: Paragraph 4: First and third sentences unclear.

Answer: Both sentences have been changed.

Reviewer #2: Discussion and Conclusions: Paragraph 6: Case of $\beta \rightarrow 0$ needs clarification. The following sentences should be reworked as well. In general, the speculation and value judgments here seem unnecessary and not directly connected to your results.

Answer: We rewrote the phrasing about $\beta \rightarrow 0$.

Reviewer #2: Discussion and Conclusions: Paragraph 7: Last sentence is unclear, where does this come from?

Answer: It has been modified to clarify its meaning.

Reviewer #2: Methods: Model equations: See previous comment on organization.

Answer: This has been fixed by including the final form of the equations of the model in the main text.

Reviewer #2: Methods: Model equations: Sentence before eqns. 3 is unclear, what do you mean by “controls access?” How this sentence transitions into the equations is difficult to parse.

Answer: The paragraph has been modified to avoid misunderstandings.

Reviewer #2: Methods: Model equations: Equations given in (3) are similar to those in another paper, correct? Why not cite this source more explicitly here, or explain in more detail? Something like “In order to model..., we follow the methods from..., where...”

Answer: We now open this subsection with an explicit reference to the original papers.

Reviewer #2: Methods: Network: Paragraph 2: You reuse k_i here, should be careful to note this (or change notation).

Answer: In this section we use the parameter k_i (lower case) which is the standard notation for each node connectivity but it was also used as the name of one of the parameters in the model equations. Along the main text we considered K_i (upper case) that describes the contextual relevance of node i in the network. In order to avoid confusion we change k_i by d_i in accordance with the degree denomination.

Reviewer #2: Methods: Network: Why not use the more conventional term “degree” to denote the number of connections of a node rather than “connectivity?”

Answer: We follow the referee advice, now we called it degree and rename the symbol to d_i

Reviewer #2: Methods: Network: Your definition of the graph Laplacian differs from convention: Wolfram gives the definition $L = D - A$, which has sign opposite yours (see <http://mathworld.wolfram.com/LaplacianMatrix.html>). Why is this?

Answer: We agree with the referee that this a way to define L . We followed the paper [33] by Mikhailov who first proposed the idea of non-localized Turing structures. In that paper, the natural generalization of the diffusive flow yields directly to our equations. in any case, both descriptions are equivalent as the apparent minus sign is absorbed by the Laplacian matrix. We explain this section in order to avoid misunderstandings.

Reviewer #2: Methods: Linear Stability Analysis: Some of the paragraph breaks here seem to be unintentional (after equations).

Answer: The whole section was carefully reviewed to avoid this.

Reviewer #2: Methods: Linear Stability Analysis: Does x^j actually depend on j ? (typo?)

Answer: It was actually a typo. In the new version of the manuscript this is fixed.

Reviewer #2: Methods: Linear Stability Analysis: After eqn. 7, as written this is not a sentence.

Answer: Fixed

Reviewer #2: Methods: Linear Stability Analysis: Before eqn. 11, what do you mean by “ J matrix?” Is this a conventional technical term (as in a Jacobian matrix), or do you mean something more arbitrary like “the matrix J ?”

Answer: Fixed, now it says “the matrix J ”

Reviewer #2: Methods: Linear Stability Analysis: The resulting eigenvalue in eqn. 13 should be explained/interpreted. I’m assuming it is meaningful because you included it, so what is important in this formula? What information does this give?

Answer: Equation 13 gives the expression to calculate the eigenvalues or growth factors. The growth factor measures the rhythm of evolution of each node to a new stationary state after a perturbation as it was said at the beginning of the paragraph prior to Eq. 13. Nevertheless, we reinforced this piece of information to avoid confusion.

Reviewer #2: Author Contributions: Details are only given for two of the four authors.

Answer: Details are now provided for all authors.

Reviewer #2: Figures: Figure 1b: See previous comment about dots and meaning of “prominent.”

Answer: This has been fixed by modifying the figure.

Reviewer #2: Figures: Figure 1c: Do error bars show 2 standard deviations from the mean?

Answer: Error bars indicate the maximum and minimum values for each group. Now this is specified in the caption.

Reviewer #2: Figures: Figure 2a is missing when it is meant to be given together with 2b and the caption, though it did show up with the separately attached figures. Perhaps this is a technical glitch.

Answer: Yes, it was a technical glitch. Thanks for noticing.

Reviewer #2: Supplementary Information: Need more description of what is given in the table. Clearly it is quantitative information, but spell out in a few sentences whether these are best fits, for which equations, etc.

The caption seems unnecessarily complicated. Why not include this information (district names) in the table itself? You could make a landscape-oriented table if necessary.

The end of the table is missing a reference.

Answer: The table has been re-written following the referee’s suggestions. Also new contents have been added to the SI following suggestions from the other referees.

Reviewer #2: Please see attached document for full review. In short, the technical results and interpretation are worthy of publication, but the prose and organization will require revision in order to clearly communicate ideas. As it stands, it is difficult to parse, especially on first read-through. After several readings I have a more thorough understanding of the work and believe it is a significant advancement in the field. I was intrigued by your results and very much enjoyed learning about your work.

Answer: We also included all suggestions included in the attached document.

REVIEWER #3:

Reviewer #3: (Remarks to the Author):

The submitted paper has the general goal of contributing to the understanding (and harnessing) of social tensions, through the study of the mechanisms underlying language shifts, with the help of some language dynamics models. This and the disappearance of languages and cultural heterogeneity are currently topics of high interest.

However, I find the paper inconsistent in various points - especially in the second model considered - and not suitable for publication.

In greater detail, the paper studies two different models which are to be considered separately.

Answer: We understand the point of view of the referee, but it was probably our fault that we did not explain it clearer in the text. What the reviewer calls first model is used just to understand the experimental data and note the existence of different temporal dynamics in the different regions. Once this is shown, we explain it by a balance between the internal complexity of each region and its interaction with the rest of the regions. For that reason, we introduced a network and thus the 'second model' as called by the reviewer. In fact, we considered a simple interaction between the different regions, by just considering a mean field approximation, i.e., each region just feels the average of all the regions. In order to avoid misunderstandings, we modified the text in various locations to make clear this point.

Reviewer #3: First Model. It features the role of internal complexity on language shift, on the base of the particular model introduced in Ref. [15]. The findings, concerning unexpected correlations between the pace of language shift and the rural/urban character of the locations, are interesting and are presented clearly enough. But they also deserve further attention and studies (and probably a more detailed discussion in a paper) and are based on one particular model (among various one proposed in the literature).

Answer: The linguistic situation in Galicia is quite unique, particularly because bilingualism is a clear option and seems to be the most stable situation. This is a fact that it is not reflected in most of the language interaction models where the final predicted evolution is always the extinction of one of the languages. This is the reason we considered this particular model that has proved to be the most adequate to describe Galicia's situation.

Despite this and following the Referee recommendation, we tried fitting the experimental data to different models. As an instance, we fitted the Galician-Spanish data to the equations in (Minett and Wang, 2008). (We refer to these as MW-model and to our equations as MP-model.) The MW equations have a similar history and motivation as ours. They have been studied analytically (we cite those studies in our paper), but they have not been compared to empirical data (to the best of our knowledge). MW-model's only stable points are such that one language attracts all the speakers always. This rules out bilingualism and language coexistence (current traits in Galicia) as long-term options. When fitting the MW-model, it consistently returned exclusive Spanish monolingualism as its steady state for every Galician region. This is at least suspicious (but definitely plausible).

Notwithstanding, we have used the MW-model to illustrate a data-driven behavior similar to the one reported in figure 1. We considered a markedly urban and a markedly rural area. We let the dynamics evolve to their steady state and then perturbed this equilibrium. We repeated this experiment with the MP-equations. The results are summarized in the figure attached below. In every case, we subtracted the value at the steady state, so that all trajectories evolve to 0. In both models, the fraction of Galician speakers tends quickly to its steady value (so this does not contribute to the converging time). Bilinguals and monolingual Spanish takes longer to relax. Consistently, the urban region is the last one to converge.

[Minett and Wand, 2008] Minett JW, Wang WSY. Modelling endangered languages: The effects of bilingualism and social structure. *Lingua* **118**, 19-45 (2008).

Thus, we can say that two different dynamics can be observed depending on whether the region is predominantly rural or urban and this trend is consistent at least in the two models considered.

We expanded this discussion in the text following the referee recommendation although we consider that a full comparison of different models is out of the scope of this paper.

Reviewer #3: Second Model. It should describe (the additional effect of) individuals moving between towns/regions, depending on what is referred to as the "contextual relevance" K_i . However, it seems to me that this model is not a simple diffusion model and for this reason it just does not fit the paper, representing something different from what the authors claim.

Answer: In fact, there was a misunderstanding along the manuscript because the model does not describe actual flows of population. The variables in the model (x_i , y_i and b_i) are percentages of population in each node speaking each one of the languages. The term taking into account the effect of the network just describes the influence of other nodes on the distribution of speakers in the original node. I.e. a node that feels from the network a large percentage of speakers of a single language is more prone to move the spectrum of its population in the same direction (but this does not mean actual displacement of the individuals).

It was our fault that this was not clearly explained in the text and we apologize. Now we introduced the appropriate changes along the text.

Reviewer #3: First, Equations (1) do not conserve the total population sizes and therefore do not describe only diffusion but also some underlying (not specified) populations dynamics. Migrations should conserve separately both x and y total populations, but neither x nor y - nor the total population - are conserved. To conserve population, one should not use coefficients K_i in Eqs. (1) but symmetrical coefficients k_{ij} (of course inside the sum over j).

Answer: This issue is connected with the previous, nevertheless it adds the question about the normalization of the variables. The model without the network term was theoretically analyzed and

it was demonstrated that the total population was kept equal to one [16]. We did not perform a similar analysis when the network term is included (and we believe it is beyond the scope on this paper) but simple simulations for multiple sets of parameters demonstrate that the total population in each node is kept equal to 1. This result has been included in the supplementary information.

Reviewer #3: First, Equations (1) do not conserve the total population sizes and therefore do not describe only diffusion but also some underlying (not specified) populations dynamics. Migrations should conserve separately both x and y total populations, but neither x nor y - nor the total population - are conserved. To conserve population, one should not use coefficients K_i in Eqs. (1) but symmetrical coefficients k_{ij} (of course inside the sum over j). A dependence of k_{ij} on both i and j would also reflect an actual contextual relevance of e.g. a city. If the diffusion flux between two towns (i) and (j) is simply proportional to the difference ($x_i - x_j$), this means really a free diffusion and therefore a contextual equivalence between the towns (a coefficient K_j and a coefficient K_j would just weight different but otherwise free diffusion processes). I do not comment on the specific results obtained from the second model, since I do not know what they really represent.

Answer: The conservation of the population was discussed in the previous point. The other issue raised here deals with the type of coupling between the nodes chosen. In fact, one can envision many different types of couplings. We choose the simplest that can incorporate the quality of urban or rural population in the easiest way. But the question is appealing, whether our results are constrained by the particular selection of the coupling or they can be considered more generic. In order to elucidate this, we considered different types of couplings following the referee recommendations. In particular, two cases were considered, one symmetric as suggested and another one that weights the contribution of each node by the amount of urban population in that particular region. As a summary, in all the cases considered a similar behavior was observed and it could be explained with the general ideas of the simpler coupling described in detail in the main text.

We consider that this discussion is relevant and now it is included in the Supplementary Information.

Reviewer #3: In conclusion, I see this paper as made up of two distinct contributions.

A first part (the first model) containing valuable results. If additional results and explanations would be provided on possible interpretations of the findings, this part may be further elaborated into a short but interesting article.

A second part, which does not seem to be consistent with the rest of the paper, which in case would need drastic revisions already at the level of model definition.

Answer: We hope that after the previous discussions, we have provided enough arguments explaining the connection between the two ‘models’. In fact, both aspects of the description are needed in order to fully understand all the richness of the system.

REVIEWERS' COMMENTS:

Reviewer #1 (Remarks to the Author):

This paper provides an innovative way of looking at sociolinguistic change and identifying the degree to which language shift differs in urban and rural spaces. It provides new insights for sociolinguists into these issues. I am unable to comment on the statistical analysis however as my expertise is qualitative.

I would recommend however that a final discussion section be included after the presentation of the analysis which draw together the key findings and recommendations for policy makers.

Reviewer #2 (Remarks to the Author):

Revisions are substantial and generally have addressed my concerns. I have a few recommendations for clarifying and copyediting the paper, but none of them prevent me from recommending publication. Please see attached text file for a detailed list, also pasted below. Line numbers reference the PDF file, attached, of the revised paper and Supplemental Information.

- Line 25: Should be "Such a social contract..."
- Line 70: Should be "Hence, we..."
- Line 76: Word "befall" isn't quite right, consider "occur" instead.
- Line 84: Word "option" isn't quite right
- Line 87: Word "focus" isn't quite right, consider "method" or similar instead.
- Line 88: Replace "along" with "over"
- Line 96: Replace "Besides" with "Additionally"
- Line 105: Replace "tells" with "describes"
- Lines 106-107: Equations 1 could be more clearly introduced. I suggest ending the sentence in line 105, then saying something like "We build upon these results, using the following equations as the foundation of our model." Then give the equations. This would adequately prepare the reader to understand why you're giving the eqns. now, without going into unnecessary detail at this point.
- Line 113: Grammar issue, could be rephrased as "allow...for the quantitative measure of..."
- Line 125: Replace "note" with "denote"
- Line 125: Replace "guesses" with "predictions"
- Line 128: Tense issue: replace "classified" with "classify"
- Line 130-134: I would remove the closing parenthesis after each group name, i.e. using A instead of A).
- Lines 157-159: Would it be simpler to explain the 5000 in a few words, i.e. "a local regulatory threshold" and point the reader to your supplemental section for more details?
- Lines 165, 167, 170: Be consistent with how you refer to equation/Eqs./equations 1.
- Line 168 and throughout: avoid use of em-dash if possible (—)
- Line 170: Replace "bilinguals" with "bilingualism"
- Line 185: Replace "previous" with "preceding"
- Line 186: Switch ordering of words "they rather" to read "...not independent, but rather, they..."
- Line 190: Omit the word "shall." Do a search and replace to remove all further instances of this word in your paper. Its use is archaic and unnecessary.
- Line 196: Omit "Thus"
- Line 203: Word "polls" isn't right
- Line 219: Sentence starting "A powerful tool..." should be reworded, to make it clear that you're using this tool to analyze the system.
- Line 224: Replace "contemplate" with "describes" or "incorporates" or similar.
- Line 234: Replace "details" with "detail"
- Line 237: Replace "slower" with "more slowly." This type of adjective/adverb issue recurs many

times in the remainder of this section, search for every instance of "slower" and "faster".

- Line 280: Replace "rather" with "instead"
- Line 292: Replace "sociolinguistic" with "sociolinguistics"
- Line 295: Replace "what" with "that"
- Line 352: Replace "equalitarian" with "egalitarian"
- Line 353: Replace "Unluckily" with "unfortunately"
- Lines 353, 357: Word "implanted" isn't quite right. Would "implemented" be better? Or "imposed" or "established" ?
- Line 406: Omit "and"
- Line 433: Replace "never" with "rarely" (unless you really mean "never"!)
- Lines 442-443: Remove trailing parenthesis at end of lines
- Lines 445, 463: Are eigenvalues negative or non-positive?
- Line 448: Reword to avoid starting sentence with an equation: "The connectivity or degree of node i is $d_i = \dots$ "
- Line 450: Replace "through" with "throughout"
- Line 453, 474: Refer to equations consistently (Eq/eq/equation)
- Line 453: Period at end of sentence.
- Line 467: Reword to make this a sentence.
- Line 470, 473, 479, 483, 486: Need punctuation at the end of the equations.
- Line 480: Replace "rhythm" with "dynamics" or "rapidity" or similar.
- Line 481: Add a word: "If the mean field..."
- Line 502: Word "elaboration" is not right.
- Line 616: Replace "p" with "b" for slope in caption of Figure 3.

- Line 11-14 of SI: Remove dashes from after section numbers
- Line 37 of SI: Last sentence must be a typo or misstatement. Your c values were experimentally found via fit to data, correct?
- Line 50 of SI: Period at end of sentence.
- Lines 52-71 of SI: Section 3 seems wholly unnecessary to me, though I read that another reviewer requested more information on normalization so I understand if it must stay.
- Lines 53, 55 of SI: Consistence in referring to equations (Eqs)
- Line 75 of SI: Section heading needs subscript for " K_i "
- Line 78 of SI: Replace "prove" with "provide evidence for"
- Line 79 of SI: Replace "above" with "in the main text" or something else more descriptive.
- Line 84 of SI: Omit second "a" to read "up to 10%".
- Line 85, 98 of SI: Be consistent with "1" versus "one"
- Line 85 of SI: Replace "constrain" with "constraint"
- Line 85 of SI: end of line should read "...integrated using Euler's method..."
- Lines 96 and 130 of SI: What do you mean by "on the right" here? The right column of sub-figures show stationary state, not the temporal evolution, right? Confusing wording, perhaps you just mean "the plots in Fig. S2 show the temporal evolution..." and similarly for S3.
- Line 144 of SI: Omit "to"
- Line 145: Replace "equalitarian" with "egalitarian"
- Line 145: Replace "experimental" with something more descriptive, as you did not perform an experiment. Perhaps "...closer to the situation at hand" or "...closer to the situation described in this paper."

Reviewer #3 (Remarks to the Author):

The reply of the authors was useful to understand better how their model represents a mean-field interaction between system and each city rather than a diffusion process and how the conservation of local populations (in each city) works in the model. With this, my main objection is removed.

However, I am left with a number of points listed below in which the text is not clear, one point

concerning the basic motivation of the paper.

If the authors can provide an explanation for the following points and possibly recheck the paper for other similar inconsistencies making the necessary corrections, the paper can be published from my side.

In general the style of the paper is not very clear and could be improved.

Line 97: Referring to the equation of Ref. [22] (paper of Abrams and Strogatz), the authors write that "The most relevant solutions to this equation (i.e. those that explain empirical data) do not allow a mixed stable population...". Notice that actually this happens for all stable equilibrium points (which are only two) of the model of Abrams and Strogatz.

Lines 92-94: "... the decline of a minority language as it comes in touch with a hegemonic one in 42 real-world cases." – sorry I do not understand what it means.

Line 102: "The AB model [24] ..." .The AB model is not that in Ref. [24], which is the Minett-Wang model; the AB model has a different dynamics and was introduced in Xavier Castello , Victor M Eguluz, Maxi San Miguel, "Ordering dynamics with two non-excluding options: bilingualism in language competition ", 2006 New J. Phys. 8 308, which is not cited.

Line 100/104: "The AS model ... has been extended in two different ways ... The AB model [24] does not allow stable mixed populations, while the model by Mira and Paredes [15] does. The later includes a third option B ..." Sorry but this are overstatements that (probably unintentionally) present the model of the authors as the only possible solution. In fact, various other 3-state competition models have been introduced, besides the model presented; and some of those 3-state models allow coexistence, either as a mixed population (e.g. the naming game with parameter beta) or as a unique bilingual population (the Baggs-Freedman model), to mention a pair of such models.

Line 175/179: "Hence, a possible explanation for the results in figure 1c is that urban areas have a greater internal complexity than rural ones – suggesting that our data is an empirical validation of the results in [35]." The authors probably mean the empirical data used, not the data of the results obtained by simulations of the first model, which make no use of networks. The authors continue by stating that: "We propose that, since the internal network structure is reflected by dynamics of varying speed, the coarse-graining of this internal complexity is captured by the different c_i values across Galician regions." This is an interesting suggestion, yet just a qualitative untested hypothesis, since the authors do not carry out any systematic comparison with Ref. [35] or with simulations of networks or any other complex social landscape representing the internal structure of a city, using their own model. They do not even try to establish a quantitative link between the c_i and the underlying (assumed) city network, only relying for the existence of such a relation on their interpretation of Ref. [35], which actually deals with a different model. In any case, the merit of such a confirmation would concern the empirical data and not the (application of) the theoretical model studied by the authors, which is the actual subject of the manuscript.

This is just to say that this hypothesis – despite reasonable – does not give any real contribution to the understanding of the topic of the relation between complexity of urban areas – which is claimed to be a main result of the paper and in fact shapes the title.

It seems to me that the manuscript is mainly an interesting application of a language dynamics model and its network extension (which could be an element of complexity), but it is not related in any way to the complexity of a social city network, which is not considered in the model.

Some additional typos:

Line 70: "Hence. we"

Line 105: "... a set of differential equations that tells the evolution..."

Compare the equations at lines 422 and 423 (in 422 should there be $P(xb)$?)

Check equations at lines 442 and 443

In addition, I find some typos here and there in the mathematical style.

- Line 25: Should be “Such a social contract...”
- Line 70: Should be “Hence, we...”
- Line 76: Word “befall” isn’t quite right, consider “occur” instead.
- Line 84: Word “option” isn’t quite right
- Line 87: Word “focus” isn’t quite right, consider “method” or similar instead.
- Line 88: Replace “along” with “over”
- Line 96: Replace “Besides” with “Additionally”
- Line 105: Replace “tells” with “describes”
- Lines 106-107: Equations 1 could be more clearly introduced. I suggest ending the sentence in line 105, then saying something like “We build upon these results, using the following equations as the foundation of our model.” Then give the equations. This would adequately prepare the reader to understand why you’re giving the eqns. now, without going into unnecessary detail at this point.
- Line 113: Grammar issue, could be rephrased as “allow...for the quantitative measure of...”
- Line 125: Replace “note” with “denote”
- Line 125: Replace “guesses” with “predictions”
- Line 128: Tense issue: replace “classified” with “classify”
- Line 130-134: I would remove the closing parenthesis after each group name, i.e. using A instead of A).
- Lines 157-159: Would it be simpler to explain the 5000 in a few words, i.e. “a local regulatory threshold” and point the reader to your supplemental section for more details?
- Lines 165, 167, 170: Be consistent with how you refer to equation/Eqs./equations 1.
- Line 168 and throughout: avoid use of em-dash if possible (—)
- Line 170: Replace “bilinguals” with “bilingualism”
- Line 185: Replace “previous” with “preceding”
- Line 186: Switch ordering of words “they rather” to read “...not independent, but rather, they...”
- Line 190: Omit the word “shall.” Do a search and replace to remove all further instances of this word in your paper. Its use is archaic and unnecessary.
- Line 196: Omit “Thus”
- Line 203: Word “polls” isn’t right
- Line 219: Sentence starting “A powerful tool...” should be reworded, to make it clear that you’re using this tool to analyze the system.
- Line 224: Replace “contemplate” with “describes” or “incorporates” or similar.
- Line 234: Replace “details” with “detail”
- Line 237: Replace “slower” with “more slowly.” This type of adjective/adverb issue recurs many times in the remainder of this section, search for every instance of “slower” and “faster”.
- Line 280: Replace “rather” with “instead”
- Line 292: Replace “sociolinguistic” with “sociolinguistics”
- Line 295: Replace “what” with “that”
- Line 352: Replace “equalitarian” with “egalitarian”
- Line 353: Replace “Unluckily” with “unfortunately”
- Lines 353, 357: Word “implanted” isn’t quite right. Would “implemented” be better? Or

“imposed” or “established” ?

- Line 406: Omit “and”
 - Line 433: Replace “never” with “rarely” (unless you really mean “never”!)
 - Lines 442-443: Remove trailing parenthesis at end of lines
 - Lines 445, 463: Are eigenvalues negative or non-positive?
 - Line 448: Reword to avoid starting sentence with an equation: “The connectivity or degree of node i is $d_i = \dots$ ”
 - Line 450: Replace “through” with “throughout”
 - Line 453, 474: Refer to equations consistently (Eq/eq/equation)
 - Line 453: Period at end of sentence.
 - Line 467: Reword to make this a sentence.
 - Line 470, 473, 479, 483, 486: Need punctuation at the end of the equations.
 - Line 480: Replace “rhythm” with “dynamics” or “rapidity” or similar.
 - Line 481: Add a word: “If the mean field...”
 - Line 502: Word “elaboration” is not right.
 - Line 616: Replace “p” with “b” for slope in caption of Figure 3.
-
- Line 11-14 of SI: Remove dashes from after section numbers
 - Line 37 of SI: Last sentence must be a typo or misstatement. Your c values were experimentally found via fit to data, correct?
 - Line 50 of SI: Period at end of sentence.
 - Lines 52-71 of SI: Section 3 seems wholly unnecessary to me, though I read that another reviewer requested more information on normalization so I understand if it must stay.
 - Lines 53, 55 of SI: Consistency in referring to equations (Eqs)
 - Line 75 of SI: Section heading needs subscript for “ K_i ”
 - Line 78 of SI: Replace “prove” with “provide evidence for”
 - Line 79 of SI: Replace “above” with “in the main text” or something else more descriptive.
 - Line 84 of SI: Omit second “a” to read “up to 10%”.
 - Line 85, 98 of SI: Be consistent with “1” versus “one”
 - Line 85 of SI: Replace “constrain” with “constraint”
 - Line 85 of SI: end of line should read “...integrated using Euler’s method...”
 - Lines 96 and 130 of SI: What do you mean by “on the right” here? The right column of sub-figures show stationary state, not the temporal evolution, right? Confusing wording, perhaps you just mean “the plots in Fig. S2 show the temporal evolution...” and similarly for S3.
 - Line 144 of SI: Omit “to”
 - Line 145: Replace “egalitarian” with “egalitarian”
 - Line 145: Replace “experimental” with something more descriptive, as you did not perform an experiment. Perhaps “...closer to the situation at hand” or “...closer to the situation described in this paper.”

Title: Internal complexity versus globalization as a coarse graining of social dynamics

Authors: Mariamo M. Juane, Luis F. Seoane, Alberto P. Muñuzuri, Jorge Mira

Reference: NCOMMS-18-14594

REVIEWER #1:

Reviewer #1: (Remarks to the Author): This paper provides an innovative way of looking at sociolinguistic change and identifying the degree to which language shift differs in urban and rural spaces. It provides new insights for sociolinguists into these issues. I am unable to comment on the statistical analysis however as my expertise is qualitative.

I would recommend however that a final discussion section be included after the presentation of the analysis which draw together the key findings and recommendations for policy makers.

Answer: A final paragraph at the end of the discussion section has been included as suggested.

REVIEWER #2:

Reviewer #2 (Remarks to the Author): Revisions are substantial and generally have addressed my concerns. I have a few recommendations for clarifying and copyediting the paper, but none of them prevent me from recommending publication. Please see attached text file for a detailed list, also pasted below. Line numbers reference the PDF file, attached, of the revised paper and Supplemental Information.

Answer: All corrections have been included. We appreciate the effort of the referee going through the whole text with such detail and thank him for it.

Reviewer #2: Line 433: Replace “never” with “rarely” (unless you really mean “never”!)

Answer: We actually mean never so we keep it the way it was.

Reviewer #2: Lines 445, 463: Are eigenvalues negative or non-positive?

Answer: They are strictly negative in our case.

REVIEWER #3:

Reviewer #3 (Remarks to the Author): The reply of the authors was useful to understand better how their model represents a mean-field interaction between system and each city rather than a diffusion process and how the conservation of local populations (in each city) works in the model. With this, may main objection is removed.

However, I am left with a number of points listed below in which the text is not clear, one point concerning the basic motivation of the paper.

If the authors can provide an explanation for the following points and possibly recheck the paper for other similar inconsistencies making the necessary corrections, the paper can be published from my side.

In general the style of the paper is not very clear and could be improved.

Answer: We have reviewed the paper and improved its style, notably thanks to the suggestions by Reviewers, and of the proof editors.

Reviewer #3: Line 97: Referring to the equation of Ref. [22] (paper of Abrams and Strogatz), the authors write that “The most relevant solutions to this equation (i.e. those that explain empirical data) do not allow a mixed stable population...”. Notice that actually this happens for all stable equilibrium points (which are only two) of the model of Abrams and Strogatz.

Lines 92-94: “... the decline of a minority language as it comes in touch with a hegemonic one in 42 real-world cases.” – sorry I do not understand what it means.

Line 102: “The AB model [24] ...”. The AB model is not that in Ref. [24], which is the Minett-Wang model; the AB model has a different dynamics and was introduced in Xavier Castello, Victor M Eguiluz, Maxi San Miguel, “Ordering dynamics with two non-excluding options: bilingualism in language competition”, 2006 New J. Phys. 8 308, which is not cited.

Line 100/104: “The AS model ... has been extended in two different ways ... The AB model [24] does not allow stable mixed populations, while the model by Mira and Paredes [15] does. The later includes a third option B ...” Sorry but this are overstatements that (probably unintentionally) present the model of the authors as the only possible solution. In fact, various other 3-state competition models have been introduced, besides the model presented; and some of those 3-state models allow coexistence, either as a mixed population (e.g. the naming game with parameter beta) or as a unique bilingual population (the Baggs-Freedman model), to mention a pair of such models.

Answer: All four previous concerns raised by the Referee were originated by the same problem, so we try to reply to all of them simultaneously. In our manuscript, we attempted to include a brief overview of the field to provide some context and to make clear that there are alternative models that could be used in this kind of research. As we shortened this brief overview in previous revisions, the writing became less clear. Some lines became obscure (e.g. lines 92-94 noted by the Referee) and others conveyed a message different to the one intended (e.g. lines 100-104, which might lead to think that our model is the only one with stable bilingualism – which is not, and we did not intend to say that). In order to fix these problems while complying with the requirement that the main text needed to be shortened, we have moved this brief literature overview to the **Methods** section and have rewritten it considerably. In the main text we still cite all the papers that we consider relevant to the field, including the work pointed out by the Referee (Xavier Castello, Victor M Eguiluz, Maxi San Miguel, “Ordering dynamics with two non-excluding options: bilingualism in language competition”, 2006 New J. Phys. 8 308), which is a relevant reference for us but had been left out in previous versions.

In more specific replies to the Referee’s concerns:

- Line 97: We did not want to leave out the possibility that other stable points might emerge for odd choices of parameters that do not represent realistic scenarios. However, that was unclear, it does not

contribute anything to the paper, and probably brought more confusion than clarity. We remove this in the newer version.

- Line 92-94: The sentence was rewritten for clarity: ‘which successfully accounts for the decline of 42 real-world minority languages in contact with hegemonic counterparts’. Note that these changes are now located in the first paragraphs of subsection **Model equations** within **Methods**.

- Line 102: We apologize for the confusion. We had in mind Minett and Wang’s model all along, while the AB model is also relevant for the field. This paragraph has been completely rewritten.

- Line 100-104: Indeed, we did not mean that our model is the only one with stable bilingualism. In the expanded discussion of the previous literature (now in Methods section) these questions are now included and clarified.

Reviewer #3: Line 175/179: “Hence, a possible explanation for the results in figure 1c is that urban areas have a greater internal complexity than rural ones – suggesting that our data is an empirical validation of the results in [35].” The authors probably mean the empirical data used, not the data of the results obtained by simulations of the first model, which make no use of networks.

The authors continue by stating that: “We propose that, since the internal network structure is reflected by dynamics of varying speed, the coarse-graining of this internal complexity is captured by the different c_i values across Galician regions.”

This is an interesting suggestion, yet just a qualitative untested hypothesis, since the authors do not carry out any systematic comparison with Ref. [35] or with simulations of networks or any other complex social landscape representing the internal structure of a city, using their own model. They do not even try to establish a quantitative link between the c_i and the underlying (assumed) city network, only relying for the existence of such a relation on their interpretation of Ref. [35], which actually deals with a different model. In any case, the merit of such a confirmation would concern the empirical data and not the (application of) the theoretical model studied by the authors, which is the actual subject of the manuscript.

This is just to say that this hypothesis – despite reasonable – does not give any real contribution to the understanding of the topic of the relation between complexity of urban areas – which is claimed to be a main result of the paper and in fact shapes the title.

It seems to me that the manuscript is mainly an interesting application of a language dynamics model and its network extension (which could be an element of complexity), but it is not related in any way to the complexity of a social city network, which is not considered in the model.

Answer:

- The Referee mentions that we do not try to establish a link between the c_i and the assumed city network. We argue that we do "try to establish a quantitative link" not with the city network, but with the average internal complexity of each Galician region. (Note that these regions are our unit of study, rather than the cities.) According to the results in [35], more complex networks (meaning, e.g., a more clustered structure were bottlenecks to the dynamics form) lead to slower convergence times towards the model's absorbing state. Note that, besides cities, there are several features that could make networks of social

interactions more or less complex. We have little access to the microscopic network structure of the Galician society for all the regions in our dataset, so we adopted a coarse-grained approach: "Cities are very likely to be more complex than other Singular Population Entities (SPEs)". And we took this in an all-or-nothing manner: "The presence of a city implies complex interactions, the presence of a rural SPE implies simple interactions". With this spirit, we made up our urbanity index u_i that treats all cities as equally complex and all rural areas as equally simple. Accordingly, regions score higher in this index the more urban population they have. Hence, we are correlating quantitatively the c_i of each region to a notion of average social internal complexity within that region. We tried other indexes, including some that make no distinction between urban and rural SPEs, but that assign more score to larger SPEs assuming that size implies more complex underlying interactions. We obtained similar results with such indexes (we do not report those, since they do not add up substantially to the paper). So, we would like to conclude by insisting, again, in that we do establish a quantitative link with the underlying internal complexity; only not at the city scale, but at the region scale – which is our scale of analysis in this paper.

- We agree with the Referee that “the merit of such a confirmation would concern the empirical data and not the (application of) the theoretical model studied by the authors”. We believe that models are limited tools to capture general qualities of the phenomena we study. Here we attempt to capture the phenomenon that certain dynamics (opinion dynamics, but probably others) evolve more slowly as the complexity of the underlying interaction network increases. This is what the authors in [35] confirm computationally for some opinion dynamics. Ideally, our results do not depend on the model. In trying to obtain more model-independent results, we tried to fit our data to other models in the literature. (Indeed, we are working towards a comparative study of existing models based on available data; but that is a completely different line of research, and very far away from completion.) Some plots resulting from those fits to another model (the Minett and Wang model) were provided in our response to the first round of review. Unluckily, what we are finding is that other models struggle to fit the data as easily as the model in our work does.

Reviewer #3: Some additional typos:

Line 70: “Hence. we”

Line 105: “... a set of differential equations that tells the evolution...”

Compare the equations at lines 422 and 423 (in 422 should there be $P(xb)$?)

Check equations at lines 442 and 443

In addition, I find some typos here and there in the mathematical style.

Answer: We have corrected these typos and many others also suggested by Referee 2 and checked consistency in the equations.